# Training Methods of Multi-Label Prediction Classifiers for Hyperspectral Remote Sensing Images

**Salma Haidar** [1,2,*] and **José Oramas** [1]

1 Department of Computer Science, University of Antwerp, imec-IDLab, Sint-Pietersvliet 7, 2000 Antwerpen, Belgium; jose.oramas@uantwerpen.be
2 Microtechnix BV, Anthonis de Jonghestraat 14a, 9100 Sint Niklaas, Belgium
* Correspondence: salma.haidar@uantwerpen.be

**Abstract:** Hyperspectral remote sensing images, with their amalgamation of spectral richness and geometric precision, encapsulate intricate, non-linear information that poses significant challenges to traditional machine learning methodologies. Deep learning techniques, recognised for their superior representation learning capabilities, exhibit enhanced proficiency in managing such intricate data. In this study, we introduce a novel approach in hyperspectral image analysis focusing on multi-label, patch-level classification, as opposed to applications in the literature concentrating predominantly on single-label, pixel-level classification for hyperspectral remote sensing images. The proposed model comprises a two-component deep learning network and employs patches of hyperspectral remote sensing scenes with reduced spatial dimensions yet with a complete spectral depth derived from the original scene. Additionally, this work explores three distinct training schemes for our network: *Iterative*, *Joint*, and *Cascade*. Empirical evidence suggests the *Joint* approach as the optimal strategy, but it requires an extensive search to ascertain the optimal weight combination of the loss constituents. The *Iterative* scheme facilitates feature sharing between the network components from the early phases of training and demonstrates superior performance with complex, multi-labelled data. Subsequent analysis reveals that models with varying architectures, when trained on patches derived and annotated per our proposed single-label sampling procedure, exhibit commendable performance.

**Keywords:** hyperspectral imaging; computer vision; deep learning; multi-label classification; two-component neural network; deep auto-encoder; patch-level classification; training schemes

## 1. Introduction

Hyperspectral imaging technology combines the power of spectroscopy with digital optical imagery. It offers the potential to explore the physical and chemical composition of depicted objects that uniquely mark their behaviour when interacting with a light source at different wavelengths of the electromagnetic spectrum. Historically, the technology was introduced in the remote sensing field [1]. However, it quickly spread to numerous other fields such as food quality and safety assessment, precision agriculture, medical diagnosis, artwork authentication, biotechnology, pharmaceuticals, defence, and home security [2], among others. Notwithstanding this richness of information, hyperspectral imaging does come with considerable challenges. Those are mostly related to the high dimensionality space of the images generated due to the presence of numerous contiguous spectral bands, the large volume of data incompatible with the limited amount of the training data available, and the high computational cost associated. Such challenges render traditional computer vision algorithms insufficient to process and analyze such images [3]. The high dimensionality problem of hyperspectral images motivated several works. In general, two types of techniques were developed in this regard, in a supervised [4] or unsupervised learning manner. These are band-selection techniques [5,6], which select the most informative subset of the bands, and feature extraction techniques [7] which transform the data to a

lower dimension. In [8], a feature extraction method called improving distribution analysis (IDA) is proposed which aims to simplify the data and the computational complexity of the HSI classification model. Ref. [9] proposes a new semi-supervised feature reduction method to improve the discriminatory performance of the algorithms. Because finding a small number of bands that can represent the hyperspectral remote sensing images is difficult, ref. [10] proposes a random projection algorithm applicable to large images that maintain class separability. Driven by this progress, hyperspectral image classification has received significant attention in recent decades leading to the development of high-performing methods. In [11], a synthesis of traditional machine learning and contemporary deep learning techniques are elucidated for the classification of hyperspectral images. However, the need for large and expensive labelled training data has constrained many deep learning methodologies to predominantly concentrate on the spectral dimension. The presence of contiguous bands induces redundancy, making the inclusion of spatial features paramount to achieving distinct separability among various classes. Several studies have leveraged deep learning to exploit both spatial and spectral contexts, enhancing the classification performance of models in hyperspectral image (HSI) classification tasks. In [12], the authors exploit deep learning techniques for the HSI classification task, proposing a method that utilises both the spatial and spectral context to enhance the performance of the models. Ref. [13] presents a joint spatial–spectral HSI classification method based on a different-scale two-stream convolutional network and a spatial enhancement strategy. Ref. [14] proposes an HSI reconstruction model based on a deep CNN to enhance the spatial features. The use of Graph Convolutional Networks (GCN) [15] and multi-level Graph Learning Networks (MGLN) [16] further underscores the diversity of approaches in addressing HSI classification. Present research endeavours predominantly concentrate on pixel-level, single-label classification, with a lesser emphasis on patch-level, multi-label classification, thereby not fully harnessing the wealth of information encapsulated in hyperspectral images. Spectral unmixing algorithms [17] have been instrumental in refining the HSI classification task by emphasising spectral variability. Several works [18,19] have focused on isolating and distinguishing multiple spectral mixes, or endmembers within individual pixels. To tackle the constraints imposed by the scarcity of labelled hyperspectral data, ref. [20] proposes two methods specifically tailored for hyperspectral analysis. These include an unsupervised data augmentation technique that performs data augmentation on the spectrum of the image and a spectral structure extraction method. Both methods aim at optimising classification accuracy and overall performance in situations characterised by few labelled samples. Concurrently, in [21], challenges arising from limited and segregated datasets are addressed by introducing a paradigm for hyperspectral image classification, which allows a classification model to be trained on one dataset and evaluated on different datasets/scenes. However, due to differences in the scenes, a gap in the categories of the different HSI datasets exists. To narrow this gap, the authors utilise label semantic representations to facilitate label association across a spectrum of diverse datasets, offering a holistic approach to hyperspectral classification amidst data limitations.

In our study, we adopt a deep learning approach in a supervised learning framework, focusing particularly on dissecting the spatial extent of images into patches of smaller dimensions. Our model is designed to preserve and exploit the joint spatial–spectral attributes, enabling the identification of multiple entities within a confined spatial area through learned features. The learning process which enables the acquisition of this knowledge is facilitated through a deep autoencoder followed by a classifier. Moreover, given the likelihood of the co-occurrence of multiple objects/entities within a depicted region, we formulate the prediction task as a multi-label prediction problem, ensuring the robustness of the model to complex scenarios encountered in hyperspectral images. Various classification networks with two components are evident in the existing literature; however, the conceptualisation of the tasks of these networks predominantly concentrate on specific aspects of data and training. Herein, data predominantly pertain to pixel-level, single-label instances. Even when utilising patches from the source images as input data, these are

densely sampled and attributed to a single label corresponding to the centre pixel of the patch to maintain the original count of the labelled pixels. The training approach for such two-component networks typically employs the *Cascade* training scheme, as used in [22]. Within the *Cascade* scheme, the feature extraction component of the network, specifically the autoencoder, is subject to independent pre-training to produce accurate reconstructions of the input. Subsequently, the decoder is substituted with a classifier which is then trained to predict the pertinent label(s), whilst freezing the weights of the pre-trained encoder.

Additionally, the literature does reflect the presence of a *Joint* training scheme as seen in [23]. However, the central objective of such studies remains primarily the analysis of the reconstruction capability of the autoencoder, with less focus on classification tasks. The *Joint* training approach implies a concurrent training process for both components. In each epoch, the autoencoder strives for a precise reconstruction from a compressed, hidden representation, whereas the classifier concurrently optimises to predict the accurate label(s) utilising the hidden representation as its input. In our research, we have performed a meticulous and systematic examination of the diverse procedures, or schemes, potentially applicable for training such a two-component network. In this context, we have delineated three distinct training methodologies for our network; namely, the *Iterative*, the *Joint*, and the *Cascade* training schemes.

The contributions of this paper are twofold. First, this paper elevates the hyperspectral image analysis by emphasising patches annotated with multi-labels, intending to resonate more profoundly with the spatial–spectral characteristics and the richness of information inherent in the images. This approach stands in stark contrast to the predominantly adhered-to practice in the existing literature of focusing on single-pixel, single-label analysis. Our observations highlight a perceptible decline in the performance of a pixel-level, multi-class classifier when subjected to training on patches annotated with single labels, corresponding to the centre pixel. It is concluded that leveraging the spatial–spectral extent available in such images, in tandem with multi-labelled ground truth, enhances the learning capability of the classifier of the latent, valuable features embedded within the hyperspectral images. Second, we systematically explore three distinct training schemes within the paradigm of multi-label prediction. The findings of our study hint at the predominance of the *Joint* scheme, supported by the attained results. However, the *Iterative* scheme shows a promising propensity for early sharing of learnable features between the feature extraction component and the classifier. This attribute of the *Iterative* scheme translates to higher performance, particularly in scenarios where data are characterised by complexity, and a mitigation in overfitting. Furthermore, our results highlight the efficacy of this scheme over the ubiquitously employed *Cascade* training scheme, especially evident in datasets marked by an increased number of samples with multi-labelled ground truth, leading to enhanced performance metrics. Those results are also observed in experiments we conducted on a patch-level, single-label classification task.

This paper is organised as follows: Section 2 positions our work with respect to existing efforts in the literature. Section 3 presents the inner workings of the proposed method and the three training schemes considered in our analysis. Those are further validated in Section 4. Finally, we put forward concluding remarks in Section 5.

## 2. Related Work

Our analysis is related to the hyperspectral patch-level, multi-label classification task employing deep models. We position our work based on closely related axes.

**Traditional machine learning as opposed to deep learning methods.** A compendium by the authors in [24] encapsulates the myriad challenges in hyperspectral image classification that cannot be addressed by traditional machine learning methodologies. Zooming in on Remote Sensing, ref. [25] provides an overview of prevalent deep learning models tailored for hyperspectral image (HSI) classification. Further, ref. [26] puts forward a comprehensive and systematic review comparing the traditional neural networks, and deep learning methods, underlining the substantial advancements made by the latter in the

realm of environmental remote sensing applications. These referenced studies primarily direct their discourse and exploration towards methodologies which facilitate pixel-level classification in a supervised context, encompassing deep belief networks, recurrent neural networks, and convolutional neural networks. In [27], the authors discuss the integration of traditional machine learning approaches with deep learning techniques by investigating the application of Deep Support Vector Machines for HSI classification. The empirical evidence compiled within these works accentuates the superiority of deep learning techniques over traditional machine learning. Our work aligns with the prevailing paradigm, focusing on the automatic and hierarchical extraction of representations from the data.

**Multi-label Prediction.** The domain of multi-label prediction in hyperspectral image analysis is relatively under-explored, especially when compared to its single-label, multi-class counterparts. Ref. [28] puts forth a method for pixel-level, multi-label HSI classification, leveraging a Stacked Denoised Autoencoder (SDAE) in conjunction with logistic regression. Their findings assert that assigning multi-labels to a pixel can elevate classification performance beyond what is achievable with a single-label approach. Another noteworthy contribution is by [29], introducing an algorithm for multi-label classification based on the fusion of label-specific features. Additionally, ref. [30] investigates a feature extraction process tailored for multi-label classification using multi-spectral images. Although this study does not extend to an empirical evaluation of hyperspectral images, it emphasises the pivotal role of multi-label classification in land cover delineation within spectral imagery.

Our work aligns with this approach, emphasising multi-label predictions in hyperspectral imagery. However, we aim to classify hyperspectral images using patches as input samples, a choice that amplifies the complexity of the task. Given the intricate nature and inherent variations in hyperspectral data, simply assigning a single label to each patch does not offer a realistic or satisfactory solution.

**Autoencoders as feature extraction method.** Autoencoders have gained recognition in several notable works [31,32], highlighted as a successful representation learning method that improves the overall performance in HSI classification tasks. Ref. [33] utilises a Stacked Denoised Autoencoder (SDAE) for feature extraction, followed by a logistic regression, fine-tuned for classification. In [34], a spectral–spatial method is proposed that modifies the traditional autoencoder through majorization minimization using multi-scale features. Furthermore, ref. [35] introduces a methodology based on multi-view deep neural networks to integrate spectral and spatial features using a small number of labelled samples.

This process begins with the extraction of spectral and spatial features using a simple deep autoencoder to reduce spectral dimensionality while preserving the spatial property of hyperspectral images. Next, it is followed by the utilisation of a multi-view deep autoencoder model, enabling the fusion of spectral and spatial features extracted from the hyperspectral image into a joint latent representation space. Subsequently, a semi-supervised graph convolutional neural network is trained on these fused latent representations to perform HSI classification. Similarly, our work aligns with these advancements by utilising a deep autoencoder for feature extraction to distil relevant information from high-dimensional data. Diverging from the aforementioned studies, we harness this information for multi-label classification.

**Patch-based input data.** The utilisation of patch-like input data samples with spatial extent smaller than the original hyperspectral remote sensing scenes is documented in the existing literature. However, our approach stands out due to its distinctive feature of annotating these patches with multi-labels. This differs from methods that assign single labels corresponding to the pixel located at the centre of the patch. In this context, ref. [36] introduces a Convolutional Neural Network (CNN) tailored for hyperspectral image data. This method extracts spectral–spatial features from the original scene. It starts from a target pixel and proceeds to densely sample neighbouring pixels to form a patch of dimensions $n \times n \times bands$. Subsequently, a single label is assigned, corresponding to the centre pixel of the designated neighbourhood/patch. This sampling mechanism

ensures the preservation of the labelled pixels inherent in the original remote sensing scene. From this process, several one-dimensional feature vectors are derived using a three-dimensional convolutional layer. These vectors are then reshaped into a two-dimensional image matrix and fed into a standard CNN for further analysis. In [37], a hybrid spectral CNN (HybridSN) combines 3D and 2D CNN layers for hyperspectral image classification. This process involves Principal Component Analysis for dimensionality reduction, followed by the segmentation of the data cube into overlapping 3D patches of size $25 \times 25$, with labels assigned based on the central pixel. In [38], the authors propose a parallel multi-input mechanism, exploiting the inherent spectral–spatial information in HSIs. It employs parallel convolution branches with varying kernel sizes (7, $5 \times 5$, $3 \times 3$, and $1 \times 1$) to extract multi-scale spatial features post-dimensionality reduction. The image is divided into neighbouring blocks or patches for this analysis. Lastly, in [39], a multi-scale 3D-CNN-based HSI classification approach is proposed called Tri-CNN. It segments data into patches and employs 3D-CNNs to extract both spectral and spatial information. The dimensionality-reduced HSI data are segmented using varying window sizes (e.g., $11 \times 11$ for the Salinas dataset and $13 \times 13$ for PaviaU). The method involves a three-branch feature fusion network, combining spatial, spectral-spatial, and spectral-only extractors. Each branch will generate a feature map. The three maps will be fused to produce the final classification results.

## 3. Materials and Methods

In this section, we present the proposed model to perform the patch-based multi-label classification.

Figure 1 illustrates the model architecture and data flow route. It is composed of two main components, an autoencoder and a classifier. Patches of reduced spatial and full spectral dimensions are sampled from the original remote sensing scene and passed into the network.

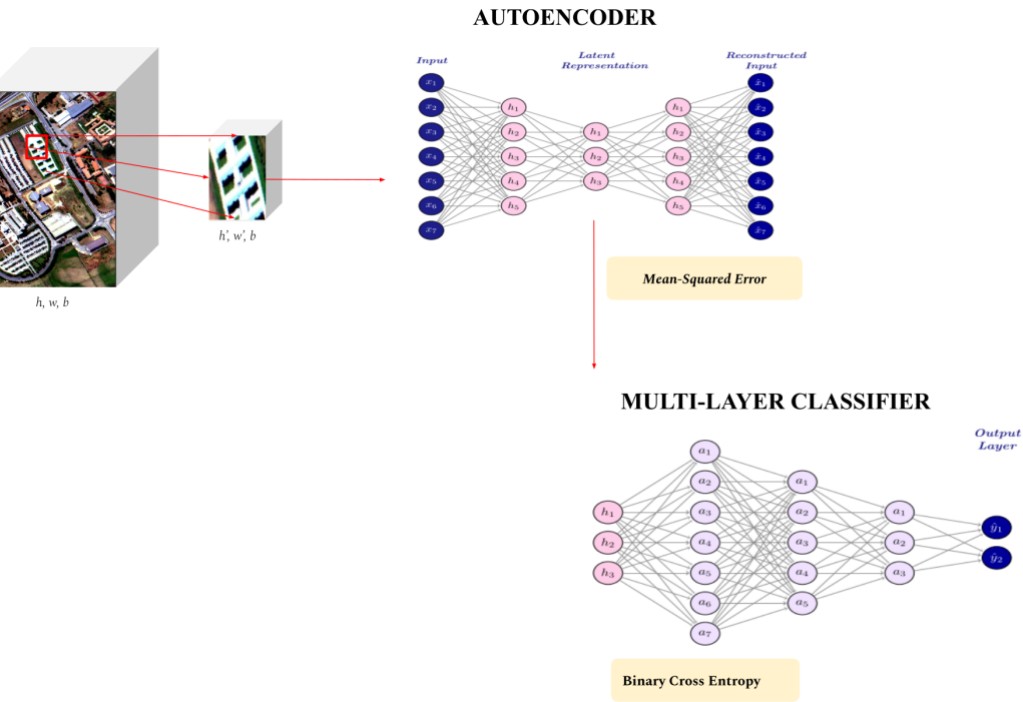

**Figure 1.** Two-componenthyperspectral image classification network composed of a deep autoencoder that feeds its hidden representation into a classifier.

Although the autoencoder reduces the dimensionality and preserves the essential features of the data to perform the reconstruction task, the classifier will highlight the

discriminatory aspects in the hidden representation of the autoencoder to identify different classes (multi-labels) present in one patch.

### 3.1. Model Architecture and Description

#### 3.1.1. Input Data

The input data, in our implementation, are structured as $X \in \mathbb{R}^{h,w,c}$, where $h, w$, and $c$ represent the height, width, and the number of spectral bands, respectively. Our methodology starts with a full scene from which we extract small patches of size $x \in \mathbb{R}^{h',w',c}$. We do that by cropping the original scene using a window of size $(3, 3)$ and applying a stride of 3 to prevent overlapping of our patches while preserving the spectral depth. Next, we annotate these patches with multiple labels representing the inherent classes within each patch. For further processing, we encode our multi-labels into one hot encoded vector where 1 indicates the presence of the classes associated with each patch. This spatial dimensionality reduction combined with the unique approach of annotation by assigning multi-labels, is a pivotal divergence from the prevalent methodologies in the literature, which predominantly assign a single label corresponding to the pixel located at the centre of the patch. It also optimises the process for handling the intricacies of multi-label hyperspectral images, an approach seldom addressed in the existing literature. Furthermore, the multi-label annotation approach is crucial in extracting more granular nuanced information from hyperspectral images (Sections 4.2 and 4.5). It offers a robust solution to address the challenges posed by the multifaceted nature of such data, thereby standing out from the existing state-of-the-art methodologies.

#### 3.1.2. Autoencoder

The autoencoder plays a crucial role in extracting and preserving essential information embedded within the spatial–spectral dimensions of the input data. The encoder, which comprises the first component of the autoencoder, receives and processes the patches before relaying them to the subsequent component or group of layers. The hidden representation of the autoencoder represents the output of the encoder. It is a compressed version of the input data encoded in the form of a vector of size $h \in \mathbb{R}^{h',w',c'}$, where $c'$ denotes the reduced number of spectral bands. This output is then passed to the decoder, the second component of the autoencoder, which undertakes the task of deconvolving the compressed data culminating in a fully reconstructed input instance. This reconstruction ensures the preservation of essential spatial–spectral information inherent in the original data.

In addition to dimensionality reduction, the autoencoder reduces redundant information by eliminating redundant neighbouring spectral bands. Those bands do not offer any additional discriminatory information, yet they contribute to the high volume/dimension of the hyperspectral data.

Equations (1) and (2) present the formal definition underpinning the functionality of the autoencoder layers.

$$h = f(W_h \times x + b_h) \tag{1}$$

Equation (1) represents the transformation of the data in each layer of the encoder component of the autoencoder. The output of the encoder component, $h$, the hidden representation or encoded data, is a compressed version of the input $x$. The matrix $W_h$ represents the weights of the encoder layer, and $b_h$ is the bias term, both optimised during the training process. The function $f(.)$, the Rectified Linear Unit (ReLU) activation function, introduces non-linearity.

$$\hat{x} = g(W_{\hat{x}} \times h + b_{\hat{x}}) \tag{2}$$

Equation (2), represents the transformation occurring in the layers of the decoder component of the autoencoder. $\hat{x}$ represents the output of the autoencoder, i.e., the reconstructed input. The matrix $W_{\hat{x}}$ and $b_{\hat{x}}$, are the weight and bias for the decoder layer, respectively, optimised during the training process. The function $g(.)$ is also a ReLU activation function,

performing the same non-linear transformation as in the encoder component, allowing the model to reconstruct the input accurately from the compressed representation $h$.

*Objective Function—Mean Squared Error (MSE).* Once the input has been reconstructed, we measure the reconstruction error. Towards that end, the Mean Squared Error (MSE) serves as our objective function, measuring the average of the squares of the errors between the reconstructed and original input.

$$MSE = \frac{1}{N} \sum_{i=1}^{N} (x_i - \hat{x}_i)^2 \tag{3}$$

where $N$ refers to the number of data points, $x_i$ is the original input data point, and $\hat{x}_i$ is the reconstructed output data point. By minimizing this loss, we optimise the reconstruction capability of the autoencoder, ensuring the preservation of essential information embedded in the spatial–spectral dimension of the input data.

### 3.1.3. Classifier

The multi-label prediction classifier is designed to learn a mapping function from an instance $x = \{x_1, x_2, \cdots, x_n\} \in X$ to a subset $l \in \mathbb{R}^c$, where c represents the entire classes in the labelled dataset. Consequently, the space of labels is defined as $y = \{0, 1\}^c$. Worth noting that in our case, the input data to the classifier are the $h \in \mathbb{R}^{h', w', c'}$ which represents the hidden representations generated by the autoencoder component of our two-component model.

The architecture of the classifier is composed of four fully connected layers. Patches are channeled through the encoder in batches, resulting in a compressed version of each patch. These compressed patches are then flattened, $h \in \mathbb{R}^{h' \times w' \times c'}$, prior to being pushed to the classifier to predict the label(s) of classes inherent in the input patch. ReLU activation function is employed along with a dropout regularisation method in each sequential layer during the training phase.

The output layer will generate logits. Predictions $\hat{y}$ are obtained by applying sigmoid function to these logits. As indicated in Equation (4), values exceeding 0.5 will indicate the presence of the corresponding classes represented at that position.

$$\hat{y} = [\hat{y}_{c_i} > 0.5] \text{ where } i \in \{1, \cdots, N\} \tag{4}$$

*Objective Function—Binary Cross Entropy (BCE).* We optimise the performance of the classifier by employing Binary Cross Entropy with Logits Loss as the objective function (see Equation (5)).

This function combines the Binary Cross Entropy (BCE) loss, employed for binary classification with a sigmoid activation, $\sigma(x) = \frac{1}{1+e^{-x}}$ as opposed to a softmax activation $\sigma(x_i) = \frac{e^{x_i}}{\sum_{j=1}^{c} e^{x_j}}$ for $i = 1, 2, \cdots, c$. BCE with sigmoid produces a probability score for each label. The computed loss is independent for each label and remains unaffected by the loss computed for another label, aligning logically with the multi-label classification where classes are not mutually exclusive.

$$l_{n,c} = -w_{n,c}[p_c y_{n,c} \times \log \sigma(x_{n,c}) + (1 - y_{n,c}) \times \log(1 - \sigma(x_{n,c}))] \tag{5}$$

Here, $c$ denotes the class number, $n$ represents the sample number in the batch and $p_c$ is the weight of the positive response for the class $c$. The loss for the $n$th sample related to the $c$th class is represented by $l_{n,c}$. In the context of multi-label prediction, this would mean the computed loss for how well the model is predicting the presence or absence of class $c$ in sample $n$.

Tables 1 and 2 illustrate the layers of each network component, offering specifics related to the shapes of the input and output of each layer. The final layer of the classifier is

designated to output the predictions, factoring in the number of classes in each dataset and the the particularities of the conducted experiment.

**Table 1.** Autoencoderarchitecture.

| Layer | Input Shape | Output Shape |
|---|---|---|
| **Encoder:** | | |
| Linear $\Rightarrow$ dropout $\Rightarrow$ relu | [1, 3, 3, bands] | [1, 3, 3, 96] |
| Linear $\Rightarrow$ dropout $\Rightarrow$ relu | [1, 3, 3, 96] | [1, 3, 3, 64] |
| Linear $\Rightarrow$ relu | [1, 3, 3, 64] | [1, 3, 3, 32] |
| **Decoder:** | | |
| Linear $\Rightarrow$ dropout $\Rightarrow$ relu | [1, 3, 3, 32] | [1, 3, 3, 64] |
| Linear $\Rightarrow$ dropout $\Rightarrow$ relu | [1, 3, 3, 64] | [1, 3, 3, 96] |
| Linear | [1, 3, 3, 96] | [1, 3, 3, bands] |

**Table 2.** Classifier architecture.

| Layer | Input Shape | Output Shape |
|---|---|---|
| **Classifier:** | | |
| Linear $\Rightarrow$ dropout $\Rightarrow$ relu | [1, 288] | [1, 3000] |
| Linear $\Rightarrow$ dropout $\Rightarrow$ relu | [1, 3000] | [1, 1512] |
| Linear $\Rightarrow$ dropout $\Rightarrow$ relu | [1, 1512] | [1, 512] |
| Linear $\Rightarrow$ dropout $\Rightarrow$ relu | [1, 512] | [1, 28] |
| Linear $\Rightarrow$ relu | [1, 28] | [1, classes] |

### 3.2. Training and Validation Process

Networks with two-component architectures conventionally utilise one of two training schemes: *Joint* or *Cascade*. This paper analyses a third scheme, the *Iterative* training scheme, while ensuring progressive and separate training of both components. This scheme allows the early sharing of features.

*Joint scheme.* In this scheme, both the autoencoder and the classifier components work in tandem to create a unified algorithm, as shown in Figure 2. This collaborative approach allows those two elements to train simultaneously. The autoencoder is responsible for generating two outputs, (1) a reconstructed version of the input, and (2) an intermediate, compressed representation, often referred to as the hidden representation. The classifier uses the latter to make predictions, specifically for multi-label output in this case. The intermediate, compressed representation is a critical component, acting as the conduit between the autoencoder and the classifier, and ensuring seamless integration and flow of information. In this scheme, the loss objective is formulated as a weighted combination of the loss objectives of both the autoencoder (Equation (3)) and the classifier (Equation (5)). Through extensive experimentation, we established that a total loss combination comprising 30% of the classifier loss and 100% of the autoencoder loss yields optimal performance. This combination was arrived at after experimenting with numerous pairs of weights, meticulously fine-tuning each to ascertain the most effective balance that maximises the efficacy of the model. During the backpropagation pass, both the autoencoder and the classifier are trained simultaneously; the autoencoder is refined to generate superior reconstructions, and the classifier is optimised to enhance the accuracy of its predictions.

*Cascade scheme.* Under this scheme (Figure 3), the autoencoder as the feature extraction component undergoes separate training to reconstruct the input, after which the trained weights are saved. Subsequently, the pre-trained autoencoder is loaded, and the encoder part is connected to the classifier. The weights of the pre-trained encoder remain frozen during this phase, allowing only the classifier to undergo training. This approach contrasts sharply with the *Iterative* scheme, where the training of each component occurs alternately. In the *Cascade* training, each training epoch involves the completion of one forward pass and one backward pass. However, during the backward propagation phase, only the

gradients related to the weights of the classifier are updated, given that the weights of the encoder part, which have been pre-trained, are frozen.

## JOINT SCHEME

Simultaneous training of the Autoencoder and the Classifier

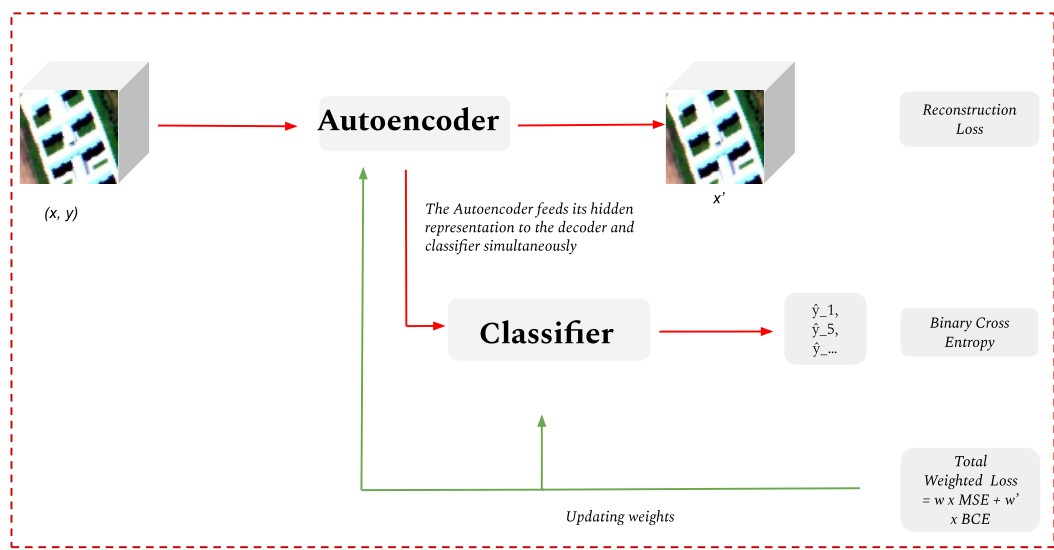

**Figure 2.** Joint training scheme.

## CASCADE SCHEME

Stage 1: Pre-training autoencoder and saving trained weights

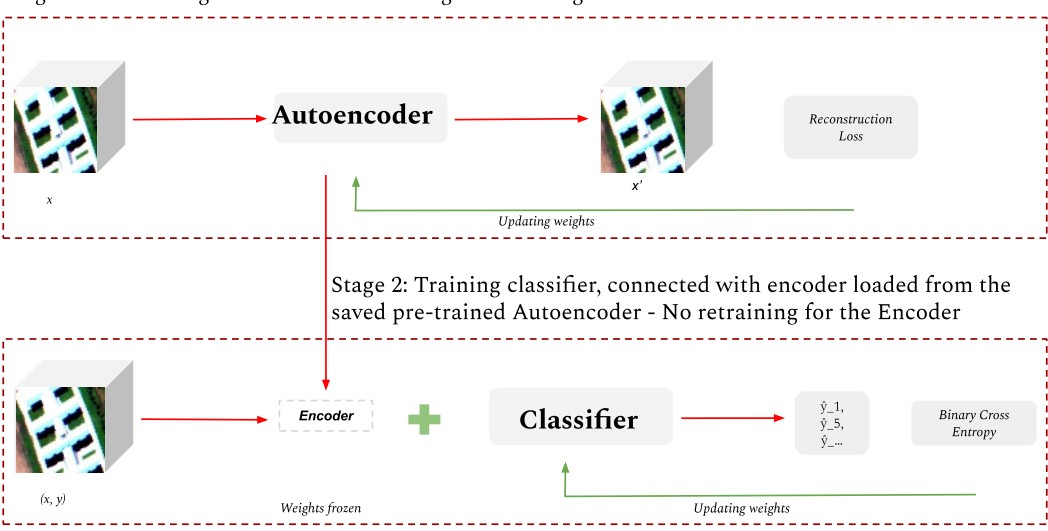

Stage 2: Training classifier, connected with encoder loaded from the saved pre-trained Autoencoder - No retraining for the Encoder

**Figure 3.** Cascade training scheme.

*Iterative scheme.* This scheme is inspired by the training process followed for Generative Adversarial Networks (GANs) [40,41]. In our implementation, Figure 4, the autoencoder and the classifier are two separate architectures, each governed by a different objective function. Training is performed in iterations, for a predetermined set of epochs, alternating between both architectures. We initialise the autoencoder, allow it to train for several epochs, and subsequently save it. Next, we initialise the classifier, load the partially trained and saved autoencoder, and pass its encoder part to the classifier. To this end, we freeze the parameters of the encoder to preclude further training. The classifier is then trained for a series of epochs. This process is repeated iteratively throughout the training phase of the network. Under this scheme, having a partially trained encoder alongside the classifier at every training step improves the performance of the classifier. Early in the training process,

the classifier will learn key features identified by the autoencoder as being informative for the reconstruction task. Leveraging those features will enhance the predictive capabilities of the classifier.

**ITERATIVE SCHEME**

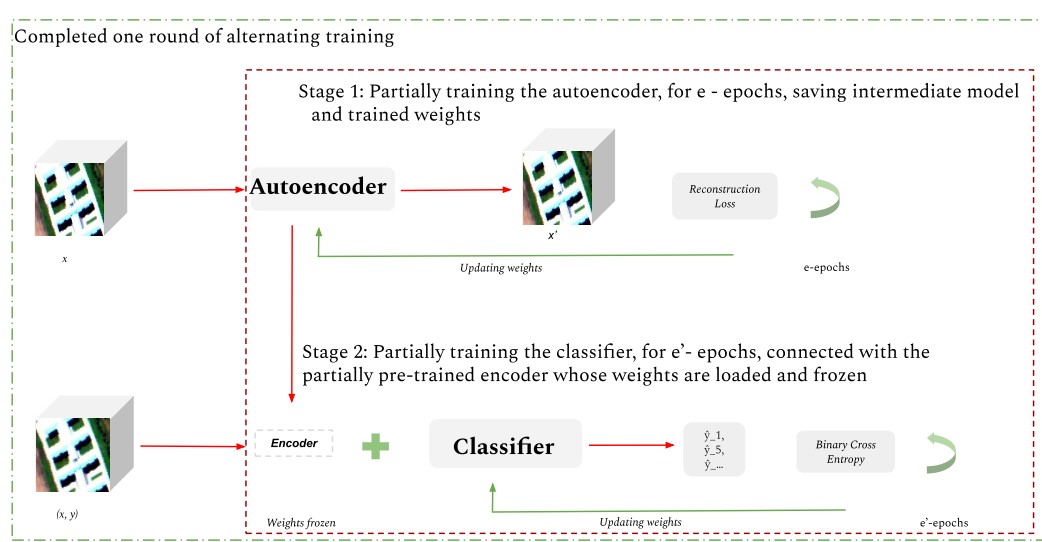

**Figure 4.** Iterative training scheme.

Under each scheme, we adopted the same architecture for the autoencoder and the classifier regarding the number of layers and input dimensions. However, we applied different hyperparameters per scheme and dataset used. Those hyperparameters were tuned to achieve the optimal performance under each setting. We incorporate L2-norm weight regularisation to the classifier loss in all three schemes. Consequently, the original loss function of the classifier, as represented by Equation (5), is extended to incorporate a regularisation term forming a new loss function.

$$l_{n,c} = -w_{n,c}[p_c y_{c,n} \times \log \sigma(x_{n,c}) + (1 - y_{n,c}) \times \log(1 - \sigma(x_{n,c}))] + \lambda \sum_{i=1}^{n} ||w_i^2|| \qquad (6)$$

In Equation (6), $\lambda$ is a scaling hyperparameter, determining the degree of penalty induced by the regularisation. The purpose of this regularisation is to control the magnitude of the weights of the model, thereby simplifying the model and mitigating the risk of overfitting. This regularisation not only helps in curtailing the complexity of the model but also contributes to enhancing its generalisation capability, ensuring more reliable and robust performance across unseen data.

### 3.3. Implementation Details

We trained, validated, and tested a multi-label prediction model on hyperspectral images of low spatial dimensions. These images have a height and width of 3 pixels, respectively, and a depth equivalent to the number of bands of the entire original scene they were sampled from. Height and width lower than $3 \times 3$ produced many patches with only one class. The larger size resulted in a reduced number of patches. We normalised the values of our data by applying *z*-score normalisation such that the mean and the standard deviation of our data are approximately 0 and 1, respectively. We applied the Adam optimisation algorithm [42].

Depending on the experiment conducted, we also selected and tuned a set of hyperparameters, such as learning rate, batch size, and drop-out rate, for training the autoencoder and the classifier. The learning rate in this context ranged from $1 \times 10^{-5}$ to $1 \times 10^{-2}$ and the batch size ranged between 100 to 240.

Both the autoencoder and the classifier utilise a learning rate scheduler, a mechanism applied during training to promote faster convergence. Learning rates are reduced at a step size corresponding to a predefined number of epochs, with a multiplicative factor denoted by $\gamma = 0.9$. Training the network generates a variable number of trainable parameters. Specifically, the autoencoder has between 35,615 and 56,108 trainable parameters. In contrast, for the classifier, trainable parameters range from 6,193,822 to 6,194,025. These observed variations stem from modifications introduced to the architectures of each component to account for intrinsic data features, particularly the number of classes and the number of spectral bands.

### 3.4. Datasets

Our method was evaluated in two publicly available datasets of remote sensing scenes [43], (Figure 5). The Pavia University Scene (PaviaU) is a hyperspectral scene acquired by the ROSIS sensor at the University of Pavia in Italy. It has 103 spectral bands ranging from $0.430\,\mu m$ to $0.86\,\mu m$ in wavelength, a spatial size of $610 \times 340$ pixels, and a geometric resolution of $1.3\,m$. The ground truth comprises nine different classes, such as trees, asphalt, and meadows, among others. Additionally, there is an undefined class labelled "background". Only 20.6% of the pixels have labels corresponding to the 9 classes, the rest are background. The Salinas Scene, a hyperspectral scene collected by the AVIRIS sensor over Salinas Valley, California. It has a spatial size of $512 \times 217$ pixels and a spatial resolution of $3.7\,m$. The original scene had 224 spectral bands ranging from $0.4\,\mu m$ to $2.5\,\mu m$ in wavelength but 20 water absorption spectral bands have been discarded. The ground truth contains 16 classes including vegetables and vineyard fields among others. Additionally, there is an undefined class labelled "background". Only 48.7% of the pixels have labels corresponding to the 16 classes, the rest are background.

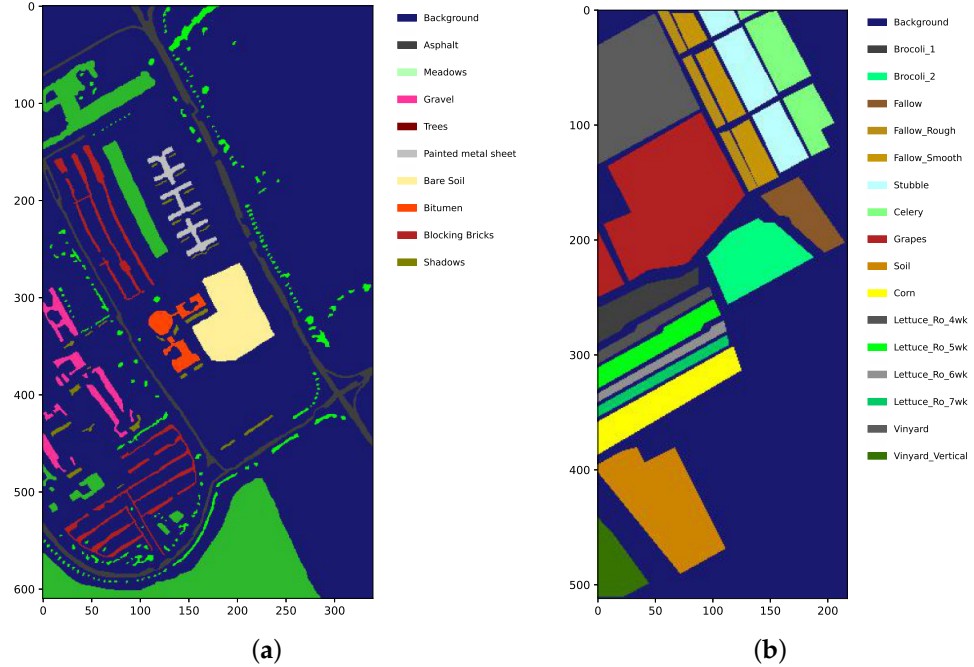

**Figure 5.** Hyperspectral remote sensing scenes: (**a**) Pavia University Scene and (**b**) Salinas Scene.

### 3.5. Patch Extraction and Label Assignment

We transformed the scenes, from the aforementioned datasets, by performing a cropping operation across the columns and rows. We have conducted that without allowing any overlapping and while preserving the original depth of the scene, i.e., the number of spectral bands. The result is low spatial dimension patches of size $(3, 3, bands)$. Next, we adopted two different approaches to assign labels to patches.

In our study, the selection of a $3 \times 3$ patch size was tailored to the specific characteristics of the PaviaU and Salinas scenes. We acknowledge, however, that the determination of the optimal patch size is intrinsically linked to various factors, such as the size of the image, spatial resolution, and the spectral complexity of the scene under study. It is important to emphasise that this choice, although suitable for our application, may not universally apply to all hyperspectral image analysis scenarios. This can be seen in domains such as terrain and urban landscape analysis, where very high-resolution hyperspectral images are acquired and used.

This underscores the necessity for a flexible approach in selecting patch sizes, tailored to the unique requirements of each application. It is imperative for future research in this field to consider these variables when determining the optimal patch size for their specific applications. Our study serves as a reference point, demonstrating the effectiveness of a $3 \times 3$ patch size for certain types of hyperspectral imagery, but it should be viewed in the context of the specific characteristics of the datasets we analysed. Although our choice of patch size was well-suited for the datasets used in our research, we admit the need for adaptability in patch size selection to suit different types of hyperspectral imagery, especially as spatial resolution and image complexity continue to evolve with advancing technology.

### 3.5.1. Multi-Label Sampling

In multi-label sampling, labels were assigned to the patches based on the classes present in each patch. Patches made up fully of pixels that belong to the background class were ignored. However, if the background class exists in a patch together with other classes, the patch is preserved and the label would include the background class. The resulting data contained a mix of patches with multi-labels and uniform patches consisting of pixels belonging to the same class.

### 3.5.2. Single-Label Sampling

In single-label sampling, patches were assigned labels corresponding to the centre pixel of the patch regardless of the classes of the surrounding pixels. In this scenario, patches where the centre pixel represented the background class were ignored. The resulting datasets are classified as either multi-labelled or single-labelled, depending on the sampling method used. However, the patches exhibit two distinct pixel distributions: (1) uniform patches, where all pixels belong to the same class, and (2) mixed patches, where pixels belong to multiple classes, including the background. Notably, for single-label sampling, the centre pixel is never of the background class.

Table 3 provides a detailed breakdown of the resulting patches and their label counts. It is crucial to note that, in alignment with our experimental framework, the background class was not ignored in our multi-label predictions. This goes in contrast with the approach taken in single-label prediction experiments where the background class was systematically excluded. Under the single-label approach, excluding the background class, facilitates the comparison with methods in the literature where the background is mostly ignored. The composition of the output layer is adapted based on the necessities of each experiment to include or omit the background class inherent in each dataset.

Table 4 provides a comparative summary of the PaviaU and Salinas datasets, focusing on the distribution of the multi-labelled patches according to the number of classes they contain and offering a clear view of the variability and complexity inherent in each dataset. For the PaviaU dataset, a significant proportion of patches contain either one or two classes. In contrast, the Salinas dataset shows a higher prevalence of patches with a single class, as seen in Table 3, suggesting a more distinct separation of classes within this dataset.

Finally, we split the data under both approaches into train, valid, and test sets adhering to approximately 80%, 10%, and 10% ratios, respectively, (Table 5). The data, model architecture, and code used in our experiments will be publicly released upon acceptance of this submission.

**Table 3.** Patches sampled from PaviaU and Salinas datasets using multi-label and single-label sampling procedures.

|  | **PaviaU** | **%** | **Salinas** | **%** |
|---|---|---|---|---|
| **Multi-Label Sampling** | | | | |
| multi-labels mixed | 3774 | 55% | 1442 | 21% |
| single-labels uniform | 3125 | 45% | 5289 | 79% |
| Total | 6899 | 100% | 6731 | 100% |
| **Single-Label Sampling** | | | | |
| single-labels mixed | 1742 | 36% | 721 | 12% |
| single-labels uniform | 3097 | 64% | 5290 | 88% |
| Total | 4839 | 100% | 6011 | 100% |

**Table 4.** Cumulative counts of patches per number of classes they contain under both PaviaU and Salinas datasets.

| Dataset | One Class Patches | Two Classes Patches | Three Classes Patches |
|---|---|---|---|
| PaviaU | 3125 | 3772 | 2 |
| Salinas | 4689 | 1466 | 16 |

**Table 5.** Train, valid, and test split of patches dataset.

|  | **Multi-Label Patches** | | | **Single-Label Patches** | | |
|---|---|---|---|---|---|---|
|  | **Train** | **Valid** | **Test** | **Train** | **Valid** | **Test** |
| PaviaU | 5588 | 621 | 690 | 3919 | 436 | 484 |
| Salinas | 5451 | 606 | 674 | 4868 | 541 | 602 |

In the realm of hyperspectral image analysis, it is indeed common to find the literature employing training percentages ranging from 1% to 30%. When applied to the two datasets in question, PaviaU and Salinas, this translates to data ranges of approximately 428 to 12,832 and 541 to 16,239, respectively. However, the appropriateness of these percentages largely hinges on the method employed, the preprocessing of the data, the available labelled data, and the computation power needed. It is pertinent to note that single-label classification, a predominant focus in this domain, utilises both the pixel-level and the patch-level analysis. Typically, in the case of patch-level analysis in the existing literature, it is predicated on densely sampling pixels from the original scene and assigning the label corresponding to the centre pixel to maintain a consistent volume of labelled data. Furthermore, in many cases, preprocessing in terms of dimensionality reduction often occurs before the training process, thus reducing the complexity of the data and the computational operations.

In our study, the decision to adopt an 80%, 10%, and 10% split for training, validation, and testing, respectively, was primarily guided by the unique characteristics of our dataset and the architecture of our method. In Sections 3.5.1 and 3.5.2, we elaborate on how our dataset was curated using multi-label and single-label sampling schemes with a stringent non-overlapping constraint. This approach significantly reduced the number of labelled patches available for use compared to the standard datasets commonly referenced in the literature. Specifically for multi-label tasks, the PaviaU and Salinas datasets comprised 6899 and 6731 patches, respectively. In contrast, for single-label classification tasks, these figures were 4839 and 6011 (Table 4). This context is crucial as it indicates that our dataset is considerably smaller—ranging between 6 and 9 times less—than those typically mentioned in the relevant literature.

Unlike plenty of work in the literature, we do not preprocess the data upfront. A common practice we notice is to apply dimensionality reduction techniques to reduce the spectral depth of the data prior to training or even prior to sampling the spatial context from the original data. This preprocessing step is part of the method we propose. In

our patch-based datasets, though small in terms of spatial context, given the $3 \times 3$ spatial window chosen, we maintained the full spectral depth and fed our cubes to the method for the dimensionality reduction to be implemented by the network. Such disparities in terms of size and the processing of the large volume of data add a layer of complexity to our methodology,

Adhering to the literature norm of 1% to 30% for training would yield an excessively small training set for our methodology, insufficient to ensure adequate generalisation in the learning process. It is crucial to underscore that our method's ability to produce the reported results, using our proposed sampling methods, adeptly addresses the inherent challenge of hyperspectral images manifested in high dimensionality coupled with the limited availability of labelled training data. Thus, the higher proportion of training data in our case is not just a methodological deviation but a necessity dictated by the specific constraints and goals of our research.

## 4. Results

### 4.1. Multi-Label Classification: Performance across Training Schemes

This experiment compares the results of the three schemes presented in Section 3.2 in the context of the task at hand, i.e., multi-label classification on hyperspectral image patches. Since our task is that of multi-label classification, we adopted the standard multi-label evaluation metrics as in [44]. More specifically, the *Accuracy*, the *Hamming Loss*, the *Precision*, and the *Recall*. According to [44], in the context of multi-label classification, the accuracy metric accounts for partial correctness, whereby the accuracy for each instance is the proportion of the predicted correct labels to the total number (predicted and actual) of labels for that instance. Overall accuracy is the average across all instances. Since we also experiment with single-label, multi-class classification, we focused our attention on *Accuracy* to perform a comparison among the different experiments we conducted.

Results: Tables 6 and 7 present an overview of the results achieved by each scheme based on the test split of the PaviaU and Salinas patches datasets. We observe that the *Cascade* scheme underperformed the other two schemes even with the Salinas dataset, which contains a significant number of uniform patches. Considering that the *Cascade* scheme is one of the commonly followed training schemes in the literature, ref. [22,33], this comes as a surprise. Moreover, *Joint* training outperformed the other two schemes in making predictions of multiple labels, mainly when tested on the PaviaU dataset. Comparing results between the two datasets, we recognise that all three schemes had a lower performance under PaviaU in making predictions of multiple labels than under Salinas. It is worth noting that the PaviaU patches dataset contains double as many instances with multiple labels as Salinas, (Table 3).

**Table 6.** Multi-label classifier: accuracy performance (in %) evaluated on multi-label patches test dataset sampled from PaviaU.

|  | *Iterative* | *Joint* | *Cascade* |
|---|---|---|---|
| Accuracy | 84.03% | 86.14% | 83.5% |
| Hamming Loss | 0.037 | 0.029 | 0.04 |
| Precision | 0.88 | 0.91 | 0.87 |
| Recall | 0.89 | 0.93 | 0.87 |

**Table 7.** Multi-label classifier: accuracy performance (in %) evaluated on multi-label patches test dataset sampled from Salinas.

|  | *Iterative* | *Joint* | *Cascade* |
|---|---|---|---|
| Accuracy | 87.61% | 86.40% | 86.47% |
| Hamming Loss | 0.015 | 0.017 | 0.02 |
| Precision | 0.89 | 0.89 | 0.88 |
| Recall | 0.93 | 0.90 | 0.92 |

*4.2. Effect of Ground-Truth Annotations on Performance*

This experiment is conducted to investigate the observations made in Section 4.1. The results from this experiment were categorised based on the number of ground-truth labels assigned to each patch in the test set. In this context, the term *multi* refers to those patches that were obtained through the *multi-label sampling* approach (Section 3.5.1), resulting in annotations that contain several labels reflecting the various classes identified within the patch. Conversely, the term *single* refers to patches that, although sampled using the same approach, contain pixels from only one class, yielding a single label in the annotation.

In this experiment, we mainly focus on classification accuracy as a performance metric, with results founded on the analysis of patches in the test sets derived from PaviaU and Salinas, respectively.

Despite previous experiments suggesting that the performance of the classifier, as gauged by average accuracy, was inferior on the PaviaU dataset patches compared to the Salinas patches, Table 8 provides a more comprehensive insight. This table breaks down the average accuracy by label type and delves into the findings presented in Tables 6 and 7.

**Table 8.** Multi-label classifier: accuracy performance (in %) based on multi-label and single-label patches.

|         | *Iterative* | | *Joint* | | *Cascade* | |
|---------|---------|---------|---------|---------|---------|---------|
|         | multi | single | multi | single | lmulti | single |
| PaviaU  | 84.75% | 83.03% | 86.29% | 85.74% | 84.31% | 83.16% |
| Salinas | 70.61% | 92.40% | 74.55% | 89.73% | 67.79% | 91.73% |

For the PaviaU dataset, which is predominantly composed of multi-label patches, all schemes display better performance on multi-label patches than on single-label patches. The *Joint* scheme consistently stands out as the best performer within this multi-label context. However, a contrasting trend is evident for the Salinas dataset. Given its characteristic of mainly having patches that are uniform (containing a single class), all three schemes achieve significantly higher accuracy on these patches. However, their performance diminishes considerably on multi-label patches.

In essence, this experiment shows that when the dataset is rich in multi-label patches, *i.e.*, mixed patches, such as PaviaU, the *Iterative* and *Joint* schemes demonstrate superior accuracy in predicting those multi-labels, albeit with a slight margin. Conversely, when the dataset leans heavily towards single-label patches, as seen in Salinas, all schemes exhibit a marked improvement in performance on the uniform patches. The *Cascade* scheme, although generally lagging behind the other two schemes, also follows this trend.

Furthermore, when examining the training and validation loss patterns, Figure 6, we find that the *Iterative* scheme converges slower to lower loss values. Nevertheless, it does not expose the model to overfitting, as is the case with the *Cascade* scheme. Even though the loss function optimised under the *Joint* training scheme is not directly comparable to the loss function optimised for the *Iterative* scheme, the former depicted better behaviour when it comes to faster convergence and lower loss values.

It is worth indicating that the total loss of the *Joint* scheme contains only 30% of the standalone classifier loss. We can also see the impact of such a contribution on the average accuracy performance of the *Joint* training in Figure 7. We observe broad and dense oscillations. Those oscillations indicate that the autoencoder has not learned the features that can help the classifier make a correct decision consistently. Examining the classifier loss component separately, we notice that it exhibits a more significant overfitting trend whose impact is diluted in the overall loss due to the low contribution of the classifier loss. If we choose another pair of weights, performance will also change, not necessarily for the better. This renders the process sensible to changes caused by other hyperparameters.

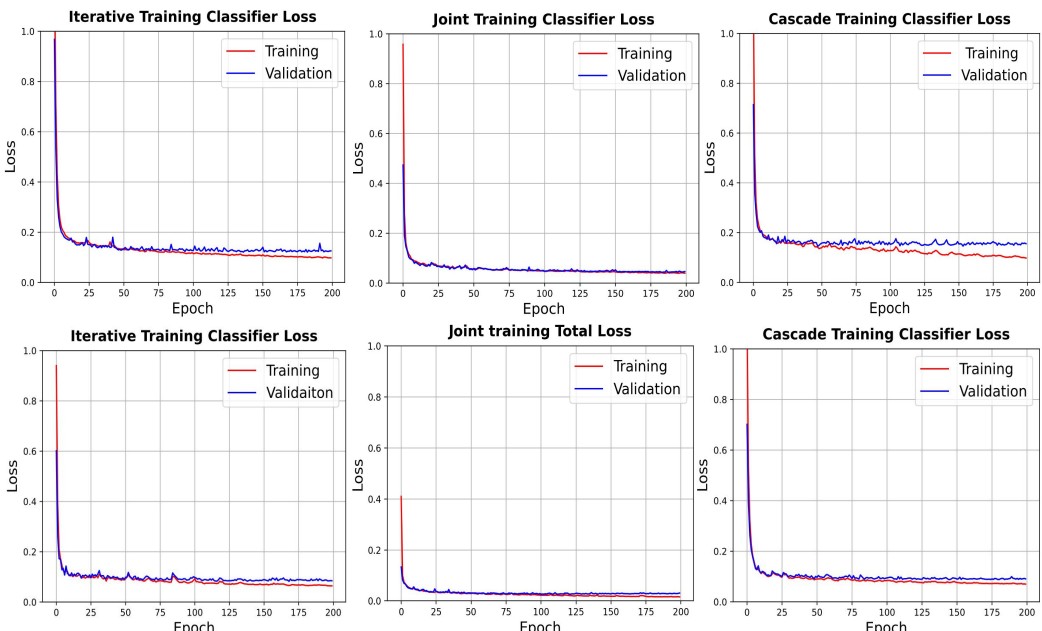

**Figure 6.** Multi-label classifier: train and valid loss under the three schemes. (**above**): PaviaU, (**below**): Salinas.

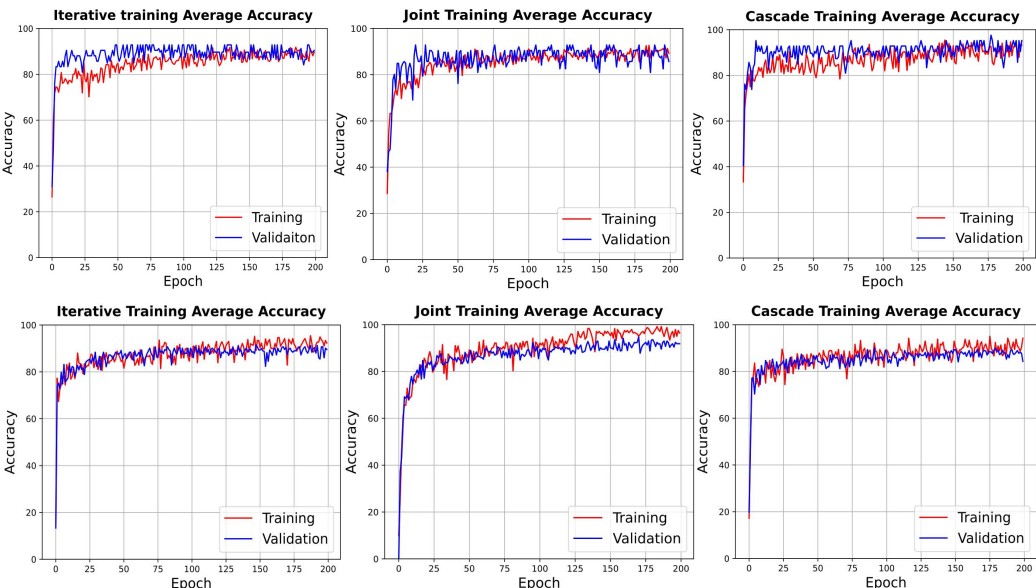

**Figure 7.** Multi-label classifier: average accuracy performance. (**above**): PaviaU, (**below**): Salinas.

### 4.3. Single-Label Classification: Performance across Training Schemes

This experiment examines the classification performance of our network under the three training schemes using single-labelled patches. We sampled those patches under the single-label sampling approach described in Section 3.4. To this end, we implemented several changes to adapt the three schemes to the new type of labels.

First, we adjusted the output layer of the classifier to reflect the number of classes after ignoring the background class. Second, we tuned a new set of hyperparameters for each method and under each dataset. Third, we adjusted the loss function to fit the new task of single-label multi-class classification. For such a task, the Cross-Entropy Loss function applies (Equation (7)).

$$l = - \sum_{c=1}^{C} y_{n,c} \log(p_{n,c}) \tag{7}$$

where $l$ is the loss value computed for a given input instance $n$. $C$ is the total number of classes in the multi-class classification problem. $c$ is the index of a specific class that ranges from 1 to $C$. $y_{n,c}$ represents the true label for class $c$ for the $n$th instance. $p_{n,c}$ represents the predicted probability that the $n$th instance belongs to class $c$. It is obtained from the output generated by the model, typically via a softmax function, ensuring the probabilities sum to 1 across all classes. Finally, for evaluation, we adopted the single-label multi-class accuracy metric calculated as the proportion of all correct predictions to the total number of data instances tested.

Under this experiment, we also applied the weight regularisation L2-norm to the classifier loss in all three schemes. Accordingly, the loss function of the classifier, as shown in Equation (7), becomes new loss function as presented in Equation (8).

$$l = - \sum_{c=1}^{C} y_{n,c} \log(p_{n,c}) + \lambda \sum_{i=1}^{n} ||w_i^2|| \tag{8}$$

Results: Table 9 presents the accuracy obtained across the different schemes per each dataset. The *Joint* scheme achieved the highest accuracy of 94.65% and 93.35% on the testing sets from PaviaU and Salinas, respectively. It was followed by the *Iterative* scheme on both datasets. Similar to the multi-label classification task, the *Cascade* scheme came third with 87.73% and 90.34%, respectively.

**Table 9.** Single-label classifier: accuracy performance (in %) evaluated on single-label patches test datasets sampled from PaviaU and Salinas.

|  | *Iterative* | *Joint* | *Cascade* |
|---|---|---|---|
| PaviaU | 90.71% | 94.65% | 87.73% |
| Salinas | 91.19% | 93.35% | 90.34% |

Figures 8 and 9 illustrate the loss and accuracy performance of each scheme on both the PaviaU and Salinas datasets. Combined with the results presented in Table 9 and despite the overall generalisation of the three schemes, the *Iterative* and *Joint* schemes exhibited overfitting and stagnation during the learning process. Additionally, this performance was obtained faster and in the earliest epochs for the *Joint* training scheme. The total loss optimisation under the *Joint* training scheme contributes to the divergence of the validation loss. The observed reduction in the performance of the classifier during validation can be attributed to higher influence given to the autoencoder in the computation of the total loss. This reduced performance indicates that the network is predominantly optimising the hidden representation for accurate reconstruction of the input data potentially neglecting the essential features useful for the classifier. The *Cascade* scheme, by contrast, converged better than the rest, yet it required a much lower learning rate $1 \times 10^{-5}$ , justifying the convergence of loss and rise in accuracy only towards the last 50 epochs of the training.

Furthermore, the *Iterative* scheme exhibited similar behaviour as the *Joint* scheme in terms of divergence of the validation loss. However, since the architecture of the *Iterative* scheme permits indirect interaction between the learned weights of the autoencoder and the classifier early on, we observe that the divergence between training and validation loss does not occur immediately and it maintains a flat level going further.

For the case of the Salinas dataset, a higher percentage of the sampled patches are uniform patches consisting of one class only. Labels, however, correspond to the centre pixel of the patch. We notice in Figures 8 and 9 that again smoothed by the weighted sum of the loss functions of the individual components (the mean squared error and the binary cross entropy), the *Joint* scheme total loss creates a smooth convergence to a low loss level. Based on the accuracy achieved employing the *Joint* scheme on the validation set, it appears that the single-label classifier was stable and consistent in capturing the learned features when making its decision at every epoch. This contrasts with the performance of the same

scheme on PaviaU and within the multi-label experiment (Figure 7). Moreover, when making its predictions, the classifier trained by the *Iterative* scheme shows a consistent learning process as reflected by the minimal oscillations of the accuracy curves in Figure 9. We link that primarily to the complex progressive nature of learning between the two parts of the network. Because uniform patches are dominant in the Salinas patches dataset, the learning process of the feature representations by the classifier seems to be easier than in the case of PaviaU.

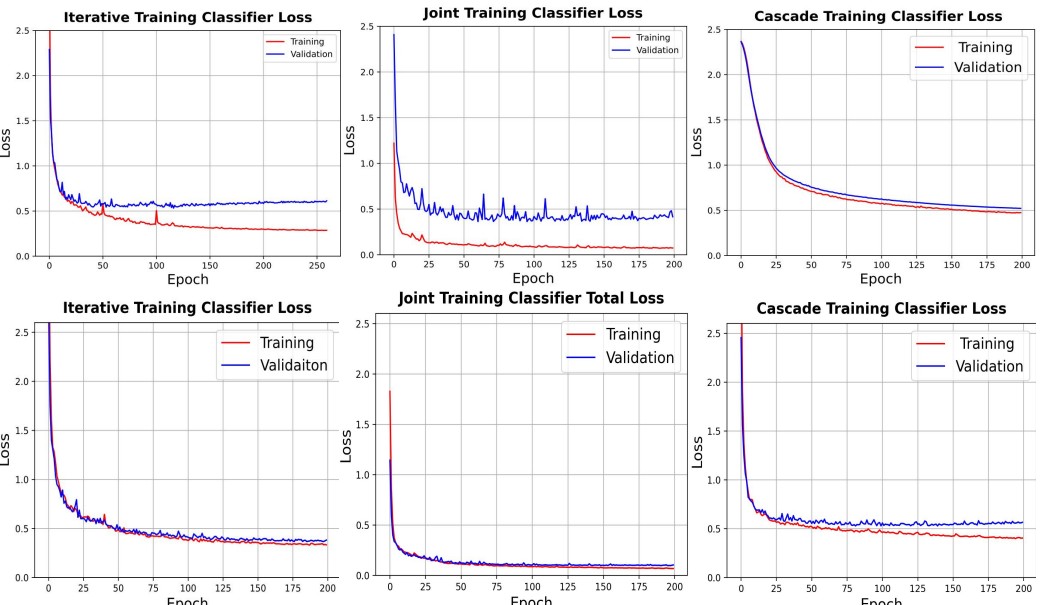

**Figure 8.** Single-label classifier: train and valid loss of the three methods. (**above**): PaviaU, (**below**): Salinas.

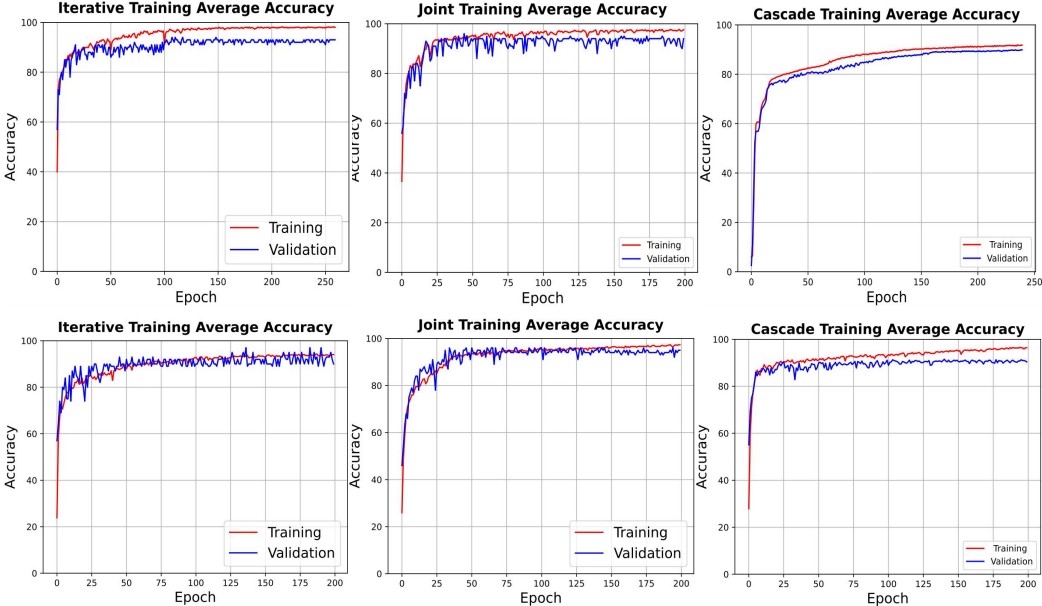

**Figure 9.** Single-label classifier: average accuracy performance. (**above**): PaviaU, (**below**): Salinas.

*4.4. Single-Label Classification: Qualitative Results*

In this section, we provide a qualitative comparison showing the predictions (in a class map format) produced by the three schemes.

Figures 10 and 11 provide a visualisation of the predicted classes under the single-label classification framework. To produce this visualisation we modified our approach to patch

generation. Initially, patches were generated with a stride of three to prevent pixel overlap. However, for this visualisation, we adjusted the stride to one, allowing pixel overlap. This adjustment aligns the number of patches with the number of labelled pixels available under both datasets, PaviaU and Salinas. We segregate the scene into three categories: (1) Background (assigned the colour black); (2) Correct Predictions (assigned the colour white); and (3) Incorrect Predictions (assigned the colour grey). Then, we pushed the full dataset of patches and collected the predicted labels. The results follow the trend we saw in our results in Table 9 where the *Joint* training scheme performed the best among the three. However, we notice a decline in the performance. Our initial results were based on a non-overlapping testing dataset. In this experiment, we opted for the overlapping approach to be able to reconstruct the original scene from the patches. This disparity in performance is primarily attributed to the variations in data volume and representation caused by the overlapping condition imposed in this experiment. Despite the fact that the data originated from the same scene and was captured by the same sensor, implying no inherent differences in the pixel data, the manner in which the patches are densely sampled leads to a distinct spatial arrangement of the classes. This variation could lead to a domain shift. Specifically, during the model's training phase, it was exposed to a certain spatial distribution of classes within the patches. However, during the inference phase, the model encounters a different distribution, which diverges from the training data. Such a discrepancy in data distribution will likely result in reduced model performance when applied to new datasets. This is because the model's learned patterns and features, optimised for the training data, may not generalise well to the newly encountered class distributions in the test data. Additionally, an analysis of the class distribution in our generated datasets of non-overlapping patches indicates the presence of an imbalance in class representation, which is inherent in the class distribution in the original data. If some classes were more abundant at the training level, they would benefit from the dense sampling, and their count would rise compared to others. This factor will exacerbate the issue as the model may not have learned to correctly classify under-represented classes due to their scarcity in the training data, contributing to increased complexity in the training process of the schemes and leading to further discrepancies in performance. Both PaviaU and Salinas share these factors. Tables 10 and 11 provide a detailed comparison of the patch datasets for PaviaU and Salinas. They show the number and percentage of patches created under the non-overlapping single-label sampling condition and compare these with the distribution of labelled samples in the original scenes.

**Table 10.** PaviaU patch dataset: non-overlapping single-label sampling of patches per class and ground truth classes for the PaviaU scene with their respective sample numbers.

| Class Name | Count (Patch Data) | % (Patch Data) | Count (Original) | % (Original) |
|---|---|---|---|---|
| Asphalt | 844 | 17.44 | 6631 | 15.50 |
| Meadows | 2048 | 42.32 | 18,649 | 43.60 |
| Gravel | 232 | 4.79 | 2099 | 4.91 |
| Trees | 339 | 7.01 | 3064 | 7.16 |
| Painted Metal Sheets | 152 | 3.14 | 1345 | 3.14 |
| Bare Soil | 561 | 11.59 | 5029 | 11.76 |
| Bitumen | 147 | 3.04 | 1330 | 3.11 |
| Self Blocking Bricks | 412 | 8.51 | 3682 | 8.61 |
| Shadows | 104 | 2.15 | 947 | 2.21 |
| Total | 4839 | 100.00 | 42,776 | 100.00 |

**Table 11.** Salinas patch dataset: non-overlapping single-label sampling of patches per class and ground truth classes for the Salinas scene with their respective sample numbers.

| Class Name | Count (Patch Data) | % (Patch Data) | Count (Original) | % (Original) |
|---|---|---|---|---|
| Brocoli_green_weeds_1 | 223 | 3.71 | 2009 | 3.71 |
| Brocoli_green_weeds_2 | 414 | 6.89 | 3726 | 6.88 |
| Fallow | 220 | 3.66 | 1976 | 3.65 |
| Fallow_rough_plow | 151 | 2.51 | 1394 | 2.58 |
| Fallow_smooth | 303 | 5.04 | 2678 | 4.95 |
| Stubble | 440 | 7.32 | 3959 | 7.31 |
| Celery | 394 | 6.55 | 3579 | 6.61 |
| Grapes_untrained | 1248 | 20.76 | 11,271 | 20.82 |
| Soil_vinyard_develop | 687 | 11.43 | 6203 | 11.46 |
| Corn_senesced_green_weeds | 365 | 6.07 | 3278 | 6.06 |
| Lettuce_romaine_4wk | 119 | 1.98 | 1068 | 1.97 |
| Lettuce_romaine_5wk | 213 | 3.54 | 1927 | 3.56 |
| Lettuce_romaine_6wk | 103 | 1.71 | 916 | 1.69 |
| Lettuce_romaine_7wk | 118 | 1.96 | 1070 | 1.98 |
| Vinyard_untrained | 816 | 13.58 | 7268 | 13.43 |
| Vinyard_vertical_trellis | 197 | 3.28 | 1807 | 3.34 |
| Total | 6011 | 100.00 | 54,129 | 100.00 |

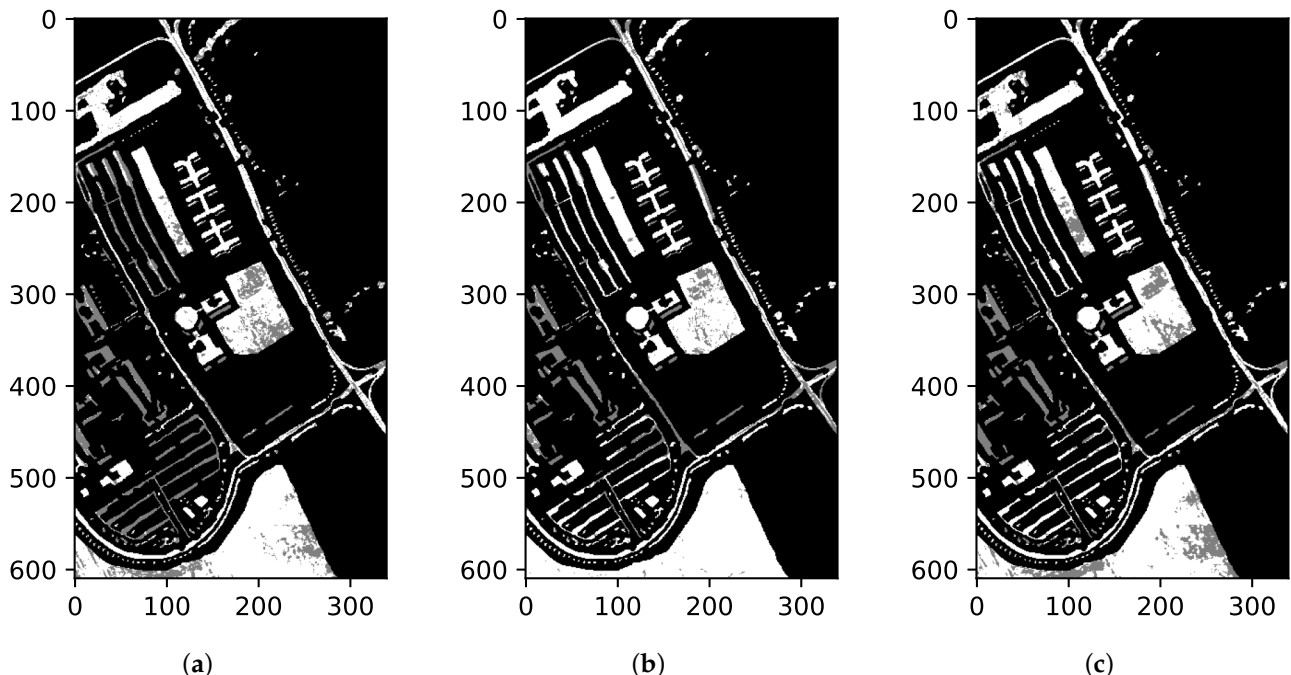

**Figure 10.** PaviaU: class map visualisation of predictions on (**a**) *Iterative Scheme*, (**b**) *Joint Scheme*, (**c**) *Cascade Scheme*.

*4.5. Single-Label Classification: Impact of Domain Shift on Performance*

To further examine the performance of the single-label classifier, we tested the classifier that was trained in Section 4.3 on two new subsets of PaviaU and Salinas. We assembled those subsets from the patches generated under the *Multi-label sampling* approach (Section 3.4) that are uniform, i.e., containing only one class as opposed to patches containing mixed classes. Consequently, we assigned the single labels of those patches based on the classes occurring in the patch and not based on the class corresponding to the centre pixel. It is worth noting that the centre pixel procedure for the label assignment characterises the data that initially trained our single-label classifier. Therefore, this experiment aims at assessing the effect of this disparity arising from training the single classifier on a combination of mixed and uniform patches, yet testing it on only uniform patches.

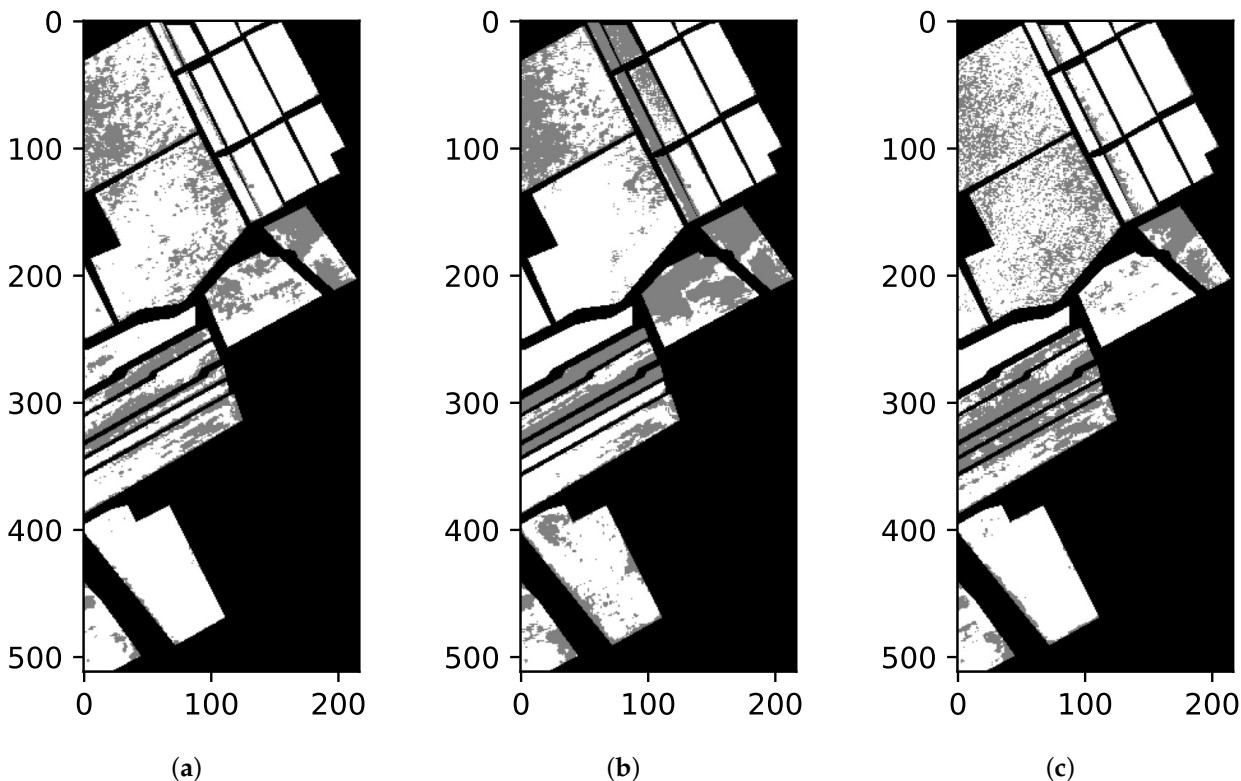

**Figure 11.** Salinas: class map visualisation of predictions on (**a**) *Iterative Scheme*, (**b**) *Joint Scheme*, (**c**) *Cascade Scheme*.

Results: Table 12 summarises the results. At first sight, the drop in performance is noticeable compared to the results reported in Table 9. For the case of the PaviaU dataset, we notice a drop of 12.31%, 2.42%, and 9.56% across the *Iterative*, *Joint*, and *Cascade* schemes, respectively. For the Salinas dataset, we notice an improvement in the performance of 1.44%, 2.60%, and 3.85% across the three schemes. These differences might be attributed to the level of disparity (domain shift) introduced by the type of patches (uniform only). As 88% of Salinas patches that our single-label classifier originally trained upon were uniform, the domain shift was relatively lower when the classifier was tested on uniform patches. Consequently, the classifier performed better on the new data and across all three schemes. The performance was reduced on the subset of the uniform patches dataset taken from PaviaU since the uniform patches generated from this scene accounted for 64% of all patches. In comparison to Salinas, this suggests a more significant domain shift. The observations above are critical if we consider that this uniform-patch setting is the simplest scenario for a single-label prediction classifier. Moreover, it shows that the common practice of assigning the centre pixel label to a patch has high costs in performance and is a practice that should be discouraged.

**Table 12.** Single-label classifier: accuracy performance (in %) on uniform single-label patches.

|  | *Iterative* | *Joint* | *Cascade* |
|---|---|---|---|
| PaviaU | 78.40% | 92.32% | 78.17% |
| Salinas | 92.63% | 95.95% | 94.19% |

### 4.6. Multi-Label versus Single-Label Classifiers: Impact of Extended Annotations on Learning Feature Representation

The goal of this experiment is to determine whether it is possible to improve representation learning by considering richer annotations in the form of multiple labels. In order to

answer this question, we compared the performance of the two models, the multi-label and the single-label classifiers. Both models were trained on patch-based datasets, however, one model restricted its predictions to a single label by using more constrained single-label annotations. Towards this goal, we modified our single-label classifier to provide multiple labels as outputs. More specifically, given an input, we selected the class labels corresponding to the top-$k$ logits. Here the number of $k$ selected logits is equivalent to the number of outputs produced by the multi-label classifier when processing the same input. It is worth indicating that this experiment is conducted using the multi-labelled data defined in Section 3.4 after removing the background class that is one of the classes included in the case of the multi-label classification. This is performed to ensure a fair comparison with the results generated from the single-label classification, which ignores the background class. Table 13 presents the average accuracy that both classifiers achieved across the three schemes; *Iterative*, *Joint*, and *Cascade*.

**Table 13.** Single-label vs. multi-label classification: accuracy performance (in %) on patches with extended annotations.

|  | Iterative Classifiers | | Joint Classifiers | | Cascade Classifiers | |
|---|---|---|---|---|---|---|
|  | multi-label | single-label | multi-label | single-label | multi-label | single-label |
| PaviaU | 90.86 | 87.07 | 94.24 | 95.76 | 91.10 | 84.32 |
| Salinas | 90.55 | 90.83 | 90.49 | 92.33 | 89.03 | 92.60 |

From the results in Table 13, we can observe that the *Joint* scheme achieved high performance under both classifiers and both datasets. However, the improvement in performance is noticeable when the results of the single-label classifier are compared to those reported under Section 4.3, Table 9. It is evident that the single-label classifier trained using the *Joint* scheme learned a representation sufficiently accurate to achieve better results when we do not limit its prediction to the highest logit generated.

Under the *Cascade* scheme, the behaviour of both the multi-label and the single-label classifiers was related to the nature of the data. Considering the multi-label context, the *Cascade* scheme performed better under PaviaU than under Salinas. Most remarkably, the multi-label classifier performance under the *Cascade* scheme was better when ignoring the background class compared to the results presented in Section 4.1, Tables 6 and 7.

When examining the results of the single-label classifier trained using the *Iterative* scheme, we notice that it achieved lower accuracy compared to the results of Section 4.3, Table 9. In other words, when the prediction of the single-label classifier in this context was not limited to the highest logit, it had lower performance. Such result indicates that the classifier trained using the *Iterative* scheme was only successful in learning the representations related to the centre pixel of the patch (the one to which the label of the patch corresponds) and failed to learn representations related to the full patch.

*4.7. Training Schemes: Accuracy per Class*

This experiment highlights how accurate each classifier is in predicting the correct class considering the three training schemes. We base our evaluation on the patches test dataset sampled from both PaviaU and Salinas scenes.

4.7.1. Single-Label Classifier

Table 14 presents the accuracy of the single-label classifier in predicting the classes of the PaviaU scene. The per-class accuracy results confirm the rankings defined based on the global performance observed in the previous experiments. The *Joint* training scheme detected 77.78% of the classes with an accuracy of 90% and above. The *Iterative* training scheme detected 55.56% of the classes with an accuracy exceeding 90% whereas the *Cascade* training scheme detected 44.45% of the classes with an accuracy exceeding 96%.

**Table 14.** Single-label classifier: per-class accuracy results, arranged in descending order of *Frequency*, measured on the PaviaU patches test dataset. The term *Frequency* denotes the count of patch instances labelled with the corresponding class.

| # | Class | Frequency | *Iterative* % | *Joint* % | *Cascade* % |
|---|---|---|---|---|---|
| 1 | Meadows | 214 | 100.00 | 98.13 | 96.73 |
| 2 | Asphalt | 84 | 78.63 | 86.33 | 80.95 |
| 3 | Soil | 46 | 90.60 | 100.00 | 76.09 |
| 4 | Brick | 43 | 83.23 | 90.70 | 88.37 |
| 5 | Trees | 33 | 95.85 | 100.00 | 100.00 |
| 6 | Gravel | 20 | 100.00 | 90.00 | 60.00 |
| 7 | Metal Sheet | 18 | 93.72 | 100.00 | 100.00 |
| 8 | Bitumen | 13 | 87.63 | 84.62 | 53.85 |
| 9 | Shadows | 13 | 74.47 | 100.00 | 100.00 |

Furthermore, Table 15 displays the performance of the single-label classifier on the Salinas dataset and across three schemes. The classifier equally predicted 100% of the correct instances of 9 out of the 16 classes available in the Salinas dataset. The *Joint* scheme eventually outperformed both the *Iterative* and the *Cascade* schemes in predicting 4 out of the remaining 8 classes positioning it at the highest rank in terms of accuracy-per-class performance among the three schemes.

**Table 15.** Single-label Classifier: Per-class accuracy results, arranged in descending order of *Frequency*, measured on the Salinas patches test dataset. The term *Frequency* denotes the count of patch instances labelled with the corresponding class.

| # | Class | *Frequency* | *Iterative* % | *Joint* % | *Cascade* % |
|---|---|---|---|---|---|
| 8 | Grapes | 119 | 73.95 | 91.60 | 71.43 |
| 15 | Vinyard | 88 | 85.23 | 72.73 | 81.82 |
| 9 | Soil | 67 | 100.00 | 100.00 | 100.00 |
| 2 | Brocoli_2 | 49 | 100.00 | 100.00 | 100.00 |
| 6 | Stubble | 45 | 100.00 | 100.00 | 100.00 |
| 7 | Celeray | 41 | 100.00 | 100.00 | 100.00 |
| 10 | Corn | 34 | 85.29 | 88.24 | 85.29 |
| 1 | Brocoli_1 | 25 | 78.63 | 86.33 | 80.95 |
| 5 | Fallow_s | 24 | 100.00 | 100.00 | 100.00 |
| 16 | Vinyard-Vert | 24 | 100.00 | 100.00 | 100.00 |
| 3 | Fallow | 25 | 96.00 | 100.00 | 96.00 |
| 12 | Lettuce_5 | 15 | 100.00 | 100.00 | 100.00 |
| 4 | Fallow_r | 17 | 100.00 | 100.00 | 100.00 |
| 11 | Lettuce_4 | 11 | 81.82 | 81.82 | 81.82 |
| 13 | Lettuce_6 | 10 | 90.00 | 100.00 | 100.00 |
| 14 | Lettuce_7 | 8 | 100.00 | 100.00 | 87.50 |

4.7.2. Multi-Label Classifier

Table 16 summarises the performance of the multi-label classifier tested on the multi-label patches test dataset sampled from PaviaU. Based on the results, the three schemes led to models with similar predictive performance. In predicting the background class, both the *Joint* and *Cascade* schemes performed slightly lower compared to the *Iterative* variant. However, the performance of the three schemes in this class was lower compared to their performance in the remaining classes. Approximately 20% of the background labels were incorrectly predicted by the classifier. This explains why in Section 4.6 we noticed the improvement in the overall performance of the multi-label classifier when we ignored the Background class. Table 17, presents the per-class performance for each scheme on the

Salinas multi-label patches test dataset. Apart from the results on the *Background* class, the three schemes have relatively close and high accuracy rates.

**Table 16.** Multi-label classifier: per-class accuracy results on the PaviaU test set, arranged in descending order of *Frequency*. The term *Frequency* denotes the occurrences of each class in the multi-labels assigned to the patches, not the count of the patches whose labels correspond to a particular class.

| # | Class | Frequency | Iterative % | Joint % | Cascade % |
|---|---|---|---|---|---|
| 0 | Background | 400 | 81.45 | 78.99 | 79.13 |
| 2 | Meadows | 237 | 95.51 | 98.55 | 96.52 |
| 1 | Asphalt | 118 | 97.54 | 97.68 | 97.54 |
| 4 | Trees | 88 | 98.70 | 98.84 | 98.70 |
| 8 | Brick | 85 | 97.54 | 97.10 | 97.10 |
| 6 | Soil | 50 | 96.52 | 99.71 | 97.54 |
| 3 | Gravel | 47 | 97.83 | 97.39 | 97.83 |
| 9 | Shadows | 26 | 99.86 | 100.00 | 100.00 |
| 7 | Bitumen | 20 | 98.41 | 98.99 | 98.84 |
| 5 | Metal Sheet | 19 | 100.00 | 100.00 | 100.00 |

**Table 17.** Multi-label classifier: per-class accuracy results on the test set, arranged in descending order of *Frequency*. The term *Frequency* denotes the occurrences of each class in the multi-labels assigned to the patches, not the count of patches whose labelled with the respective class.

| # | Class | Frequency | Iterative | Joint | Cascade |
|---|---|---|---|---|---|
| 0 | Background | 148 | 87.39 | 87.39 | 85.02 |
| 8 | Grapes | 140 | 93.77 | 93.77 | 92.58 |
| 9 | Soil | 82 | 99.70 | 99.70 | 99.41 |
| 15 | Vinyard | 83 | 93.92 | 94.21 | 93.03 |
| 2 | Brocoli_2 | 49 | 100.00 | 100.00 | 100.00 |
| 6 | Stubble | 48 | 99.85 | 99.85 | 99.85 |
| 10 | Corn | 42 | 98.22 | 98.81 | 98.67 |
| 7 | Celeray | 35 | 99.70 | 99.56 | 99.85 |
| 5 | Fallow_s | 37 | 99.56 | 99.56 | 99.70 |
| 1 | Brocoli_1 | 26 | 100.00 | 99.85 | 100.00 |
| 12 | Lettuce_5 | 22 | 99.70 | 99.41 | 99.41 |
| 3 | Fallow | 22 | 99.70 | 99.85 | 100.00 |
| 14 | Lettuce_7 | 19 | 99.56 | 99.56 | 99.70 |
| 11 | Lettuce_4 | 18 | 99.70 | 100.00 | 99.26 |
| 4 | Fallow_r | 20 | 99.85 | 99.70 | 99.70 |
| 16 | Vinyard-Vert | 16 | 99.85 | 100.00 | 100.00 |
| 13 | Lettuce_6 | 15 | 99.85 | 99.85 | 99.85 |

### 4.8. Comparison with Existing Work

In this section, we quantitatively compare the performance obtained in our study with two related methods from the literature. This is conducted on two fronts. First, at the data level, we trained and tested the HSI-CNN method developed in [36] using the single-label patches dataset we sampled. Second, at the training scheme level, we trained in a joint manner and tested the two-branch autoencoder method (TBAE) developed in [23] using the patches dataset and compared the performance with that of our two-component network, (Section 3).

#### 4.8.1. Single-Label Classifier Performance: Training Different Architectures Using Patches Dataset

The experiment investigates whether the performance of the single-label classifier trained with a smaller number of patches would differ given the different underlying architecture employed for learning representations.

In [36], the classification method is based on a convolutional neural network to perform the hyperspectral image classification task, whereas our network adopts an architecture of fully connected layers. Similar to our work, the HSI-CNN method is trained using patches of $n \times n \times bands$ extracted from the original remote sensing scene to preserve the spatial–spectral aspect of the data. In addition, assigning labels to patches is based on the label corresponding to the centre pixel, a protocol we also use for our single-labelled patches. There is, however, one point to note. The sampling process originally followed in [36] differs from ours. They perform a dense sampling that maintains the same count of labels of the original remote sensing scene, whereas in sampling our patches we ensured that no overlapping between the patches is allowed. This reduces the volume of the dataset and the count of the labels. By doing so we reduce the computational time and the required resources to conduct the experiments.

We based our experiment on the implementation of the HSI-CNN method provided by [11] in their DeepHyperX toolbox (https://github.com/nshaud/DeepHyperX, (accessed on 23 December 2022)) [45]. Given that the volume of the input data and the process of sampling are different, training the model required tuning a set of hyperparameters that is not completely identical. Similar to the DeepHyperX implementation, we used the stochastic gradient descent (SGD) optimiser, a cross-entropy loss function and a learning rate scheduler. Moreover, opposite to the mentioned implementation the scheduler differed from one dataset to the other. For the PaviaU dataset, we used the step-learning rate scheduler which enforces a reduction in the learning rate value at the end of a predefined set of epochs. This learning rate adaptation, however, did not work for the Salinas dataset, where we applied the reduce-on-plateau scheduler. When the model was trained on patches extracted from the Salinas dataset, empirical evidence suggested that the learning rate reduction pattern was not correctly captured using a predefined number of epochs. In contrast, the reduced plateau scheduler allowed the model to automatically adjust the learning rate when the validation loss performance stopped improving rather than enforcing a fixed period to apply the change. All details related to the hyperparameters can be found in the Supplementary Material.

Table 18 contrasts the results of training the CNN-HSI method using the non-overlapping patches dataset sampled in our work against the results achieved by the DeepHyperX implementation of the same method using the dense sampling process of patches proposed by [36].

**Table 18.** Single-label classification: accuracy performance (in %) of the two-component network under the three training schemes, the HSI-CNN method trained using the non-overlapping patches dataset, the HSI-CNN method trained using densely sampled patches and the TBAE method trained using the non-overlapping patches dataset.

|         | *Iterative* | *Joint* | *Cascade* | **HSI-CNN** **Non-Overlapping** | **HSI-CNN** **Dense** | **TBAE** |
|---------|-------------|---------|-----------|---------------------------------|-----------------------|----------|
| Pavia   | 90.71       | 94.65   | 87.73     | 70.56                           | 73.05                 | 89.56    |
| Salinas | 91.19       | 93.35   | 90.34     | 87.33                           | 69.05                 | 91.96    |

Figures 12 and 13 contrast the performance of the DeepHyperX implementation against our implementation of the HSI-CNN method using the dataset of patches that we originally used to train our own two-component network. Both our results and those achieved by the DeepHyperX implementation could not match those reported in [36]. This is not surprising, given the fact that we are using a different dataset and a different set of hyperparameters. Nevertheless, we did observe one similar behaviour in our implementation of the results reported in the original paper: the progressive upward sloping of the accuracy curves.

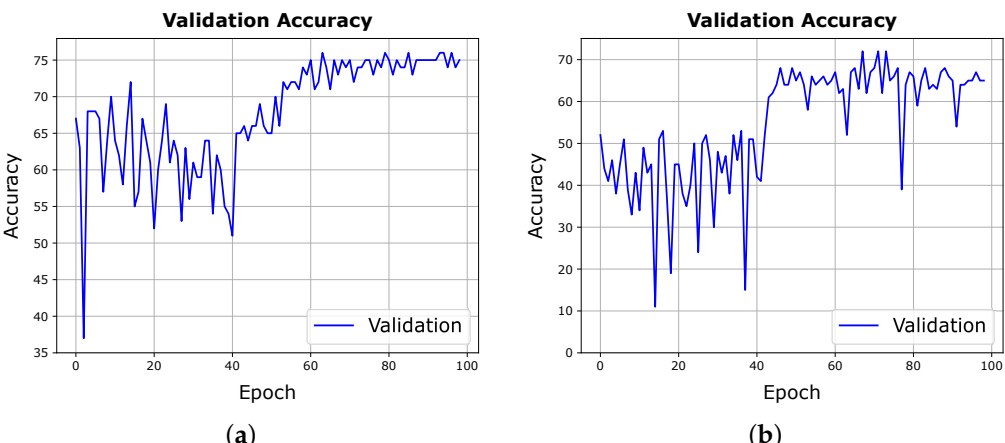

**Figure 12.** Validation accuracy performance of the HSI-CNN method developed by [36] and implemented by [11] using densely sampled patches from the remote sensing scenes. (**a**) PaviaU, (**b**) Salinas.

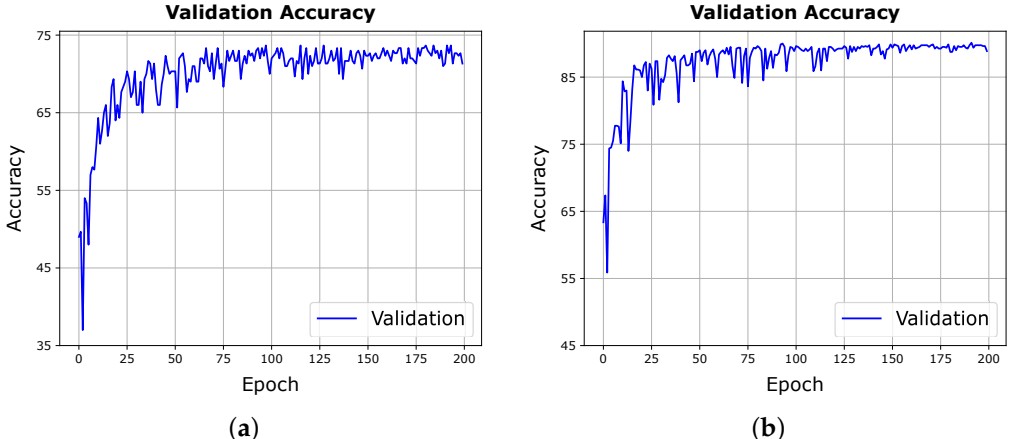

**Figure 13.** Validation accuracy performance of the method developed by [36] and implemented by [11] that we amended to use our sampled non-overlapping patches dataset. (**a**) PaviaU, (**b**) Salinas.

In summary, the high correlation between a pixel and its neighbouring pixels, which share common characteristics, can be captured when a model is trained using patches datasets, as opposed to spectral-only pixels. The nature of patches can provide an option to overcome the shortcoming of having limited labelled datasets. Hence, in the absence of large labelled samples of remote sensing datasets, deep architectures like the one presented in [36] can be easily and more efficiently trained on a smaller-sized dataset. One that preserves both the spatial and the spectral dimensions and leverages the correlation between the neighbouring pixels.

4.8.2. Single-Label Classifier Performance: *Joint* Training Scheme

Ref. [23] presents a semi-supervised method that extracts features from the unlabelled data by training a single-layer autoencoder whose hidden layer inputs into a classifier with a softmax layer. Simultaneously, the encoder and the classifier exploit the labelled data to perform a classification task. In their analysis, they show the possibility of jointly training the two-branch autoencoder (TBAE) model using a limited number of unlabelled and labelled pixels of the remote sensing scenes. Their work appears to share structural similarity with our work in terms of the two-component architecture encompassing an autoencoder and a classifier and combined with the *Joint* training scheme. Based on that, we reproduced their method and allowed the training to proceed using our single-label patches dataset, sampled from PaviaU and Salinas in accordance to the *Single-label sampling* approach described in Section 3.4.

Figure 14 exhibits the results of the training conducted. Even with the shallowness of the architecture, the joint training of the model allowed good performance in terms of validation accuracy. The latter achieved on average 89.56% on the PaviaU patches dataset and 91.96% on the Salinas patches dataset (Table 18). However, this performance fell short when compared with the results achieved by our architecture (Section 3) trained using the *Joint* scheme which was 94.65% and 93.35% on PaviaU and Salinas, respectively, (Section 4.3, Table 9).

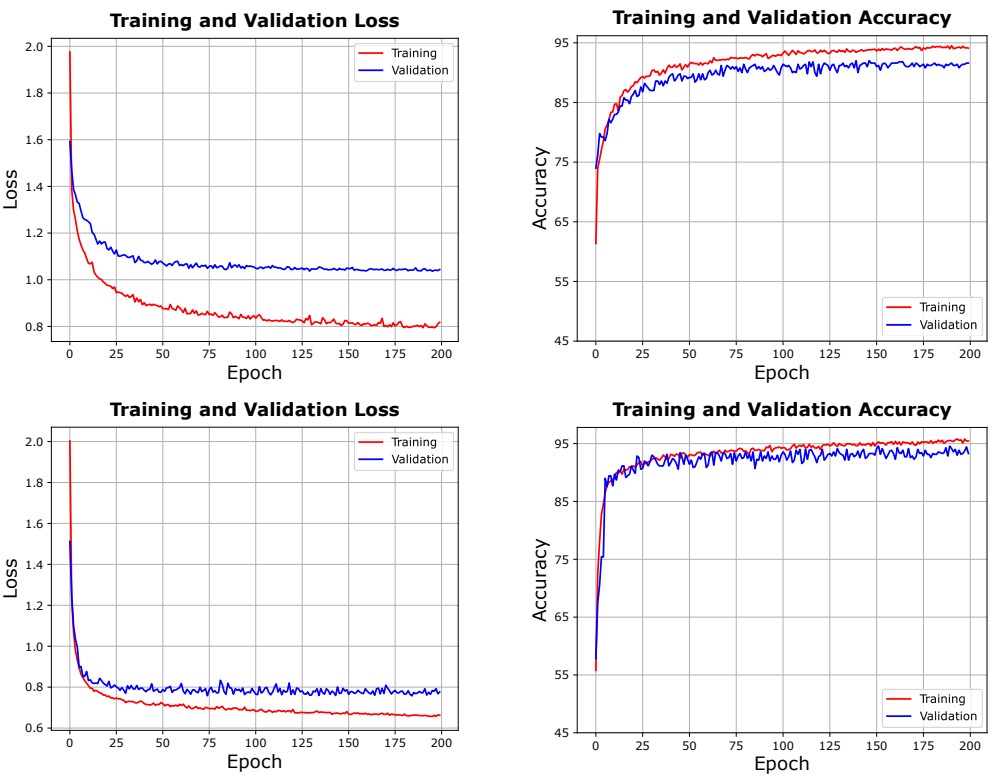

**Figure 14.** Performance in terms of loss and accuracy on the method developed by [23] using non-overlapping sampled patches. (**above**) PaviaU, (**below**) Salinas.

In our opinion, this is justified for two main reasons. First, the deeper architecture of our two-component network allows the encoding of a richer representation. Second, the different shape of the input data, i.e., the patches, which preserves the joint spatial–spectral aspect of the original remote sensing scene. As opposed to the method from [23], which was trained using pixels containing only spectral information. Furthermore, the results demonstrate that our architecture is capable of learning intrinsic feature representations embedded in the patches despite their limited volume.

## 5. Conclusions

In this paper, we developed a two-component network, consisting of an autoencoder and a classifier, to perform a multi-label hyperspectral classification task. Diverging from the conventional approach of utilising single-label pixel-level input data, we opted for patches extracted from the hyperspectral remote sensing scenes. This approach leverages the abundance of information and the intrinsic correlation among the pixels present in those scenes. We rigorously trained our network and evaluated the performance of the classifier, focusing on the assignment of multi-labels to patches instead of single labels. Beyond the commonly used *Joint* and *Cascade* training schemes found in the literature for two-component networks, we investigated the *Iterative* training scheme. Our evaluations, which spanned two datasets and classification tasks (multi-label and multi-class classifications), indicate that the *Cascade* scheme performs the least, whereas the *Joint* scheme stood out as the most proficient across all scenarios. Nevertheless, it has the drawback of requiring an

expensive parameter search procedure to determine the optimal weight combination of the constituents of the total loss, risking overfitting. Such a combination might be optimal given a specific data split and selected hyperparameters, but its generalisability under varying conditions and hyperparameters is uncertain.

According to our experiments, the *Iterative* scheme, with its progressive learning approach, permits the sharing of features between the two parts of the network from the early stages of training. This scheme obviates the requirement for an intricate search for specific hyperparameters that are not evident or principled and consistently yields good results. This is noticeable, particularly with complex datasets containing numerous multi-label patches that preserve the spatial and spectral characteristics of hyperspectral images. Moreover, our results suggest that the common practice of assigning a label corresponding to the centre pixel of a patch has high costs in predictive performance and should be discouraged. Furthermore, our findings reveal that architectures, fundamentally different in nature and deeper than our two-component network, could perform well when trained on datasets smaller in volume yet more abundant in spatial–spectral information; namely, the patches dataset used in our experiments. Our observations also showed enhanced performance when employing the *Joint* scheme to train our two-component architecture with patches, compared to training a shallower architecture with a small subset of labelled pixels containing only spectral information. For future work, our study could be extended by further validating the schemes and methods on other remote sensing datasets and possibly other types of hyperspectral images. Recent learning techniques; such as self-supervised learning, and modern architectures; such as transformers, will be considered to further boost the performance of our classifier.

**Supplementary Materials:** The following supporting information can be downloaded at: https://www.mdpi.com/article/10.3390/rs15245656/s1.

**Author Contributions:** Conceptualization, S.H. and J.O.; Methodology, S.H. and J.O.; Software, S.H.; Validation, S.H.; Formal analysis, S.H. and J.O.; Investigation, S.H.; Resources, S.H. and J.O.; Data curation, S.H.; Writing—original draft, S.H.; Writing—review & editing, S.H. and J.O.; Visualization, S.H.; Supervision, J.O. All authors have read and agreed to the published version of the manuscript.

**Funding:** The research presented in this article is part of the project "Learning-based representations for the automation of hyperspectral microscopic imaging and predictive maintenance" funded by the Flanders Innovation & Entrepreneurship—VLAIO, under grant number HBC.2020.2266.

**Data Availability Statement:** The data presented in this study are available on request from the corresponding author. The data are not publicly available due to size limitations.

**Conflicts of Interest:** Author Salma Haidar was employed by the company Microtechnix BV. The remaining authors declare that the research was conducted in the absence of any commercial or financial relationships that could be construed as a potential conflict of interest.

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
