# Peer review of "Training Methods of Multi-Label Prediction Classifiers for Hyperspectral Remote Sensing Images"

_remotesensing, doi:10.3390/rs15245656_

Round 1

Reviewer 1 Report

Comments and Suggestions for Authors

In this manuscript, authors had presented a novel two-component deep learning network for multi-label and patch-level classification of hyperspectral remote sensing images. The deep learning network achieved a better performance in two hyperspectral remote sensing datasets. Moreover, authors systematically compared the effects of three distinct training schemes on the network's prediction performance. This study is a meaningful topic for hyperspectral remote sensing images classification. I think the manuscript is appropriate for publication in this journal. However, there are several issues should be addressed before acceptance:

1.       There were some spellings wrongs in the manuscript, such as, “mayrid” should be “myriad” in Page 3 Line 127; “exits” should be “exist” in Page 11 Line 372.

2.       I think the “endmembers” dose not need to be italicized in Page 2 Line 59.

3.       I think the equation 5 (the loss function of classifier net) could be typeset on one line.

4.       In this manuscript hyperspectral image and hyperspectral imaging used same abbreviation HSI”. It may cause confusing.

5.       I think the time cost of network’s predictions could be supplemented.

Comments on the Quality of English Language

None

Reviewer 2 Report

Comments and Suggestions for Authors

The paper introduces a new approach to perform HSI classification task using multi-label classifier on HSI patches. The approach consists of 2 components, an autoencoder and a classifier. This proposed network was trained with different training schemes; namely iterative, joint and cascade.

Results show the iterative training permits the sharing of features between the two parts of the network from the early stages of training. Joint scheme showed enhanced performance compared to training a shallower architecture with a small subset of labelled pixels containing only spectral information.

I have few concerns in the results section:

1. line 391: the training percentage was set 80%, which is very high in my opinion. Many papers in literature use values from 1% to 30%. I understand from the nature of the problem this number should be high. what will be the advantage of this approach if I need vary large volume of data for training.

2. Section 4.7.1: when comparing to single label classifiers, the authors compared against an old algorithm was published in 2018. many algorithms were published since then and their implementations are available in github, for example (just few to mention):
     a. (2020) https://github.com/gokriznastic/HybridSN
     b. (2023) https://github.com/mqalkhatib/Tri-CNN
     c. (2022) https://github.com/Pl-2000/PMI-CNN
 I suggest to compare the proposed model against the above models to check the model performance.

3. Since the paper is on HSI classification, I'm interested to see qualitative comparisons by showing the class maps of each model. However, the paper didn't show any.

Reviewer 3 Report

Comments and Suggestions for Authors

The authors present a two-component methodology for classifying hyperspectral images that utilizes an autoencoder and a multi-layer classifier. The primary novelty of the work is the focus on multi-label, patch-level classification, which is understudied in the literature related to classification of hyperspectral images, despite the fact that it's very typical for a hyperspectral image pixel to contain spectral information from multiple classes for applications spanning microscopy to satellite imagery.  The authors present a thorough description of the proposed model architecture and assess the results on two publicly available and well documented hyperspectral image data sets. training schemes single vs. multiclass labels.  The authors present a comprehensive discussion of the results for several important comparisons.  In general I found the paper to be well-written and the claims are well-substantiated by the data presented.  The successful strategy of non-overlapping patches for training data is quite interesting. The work advances the field of hyperspectral image analysis and should be of interest to the readers in Remote Sensing. Specific suggestions for improvement are noted below.

- Additional details regarding the multiclass labels would be helpful.  When the authors assign multiple class labels for each patch, how many labels are typically assigned?  It is implied that this number is >2, but no further details are given. Is it anywhere from 1-9 for each patch or is the value typically much lower? Were these class labels selected based on the ground truth pixel assignments in the data sets?

- The strategy to remove patches containing all background pixels is interesting and has positive implications in many applications of hyperspectral imaging, including microscopic cell and tissue imaging where the pixels of interest are often sparse relative to the image scene.   It seems the ideal performance would be observed where the number of patches (or pixels) assigned to any one label were equal, thus preventing bias.  Can the authors provide some insight on whether this is true?

- Optimal patch size is dependent on the image size and the spatial resolution and spectral complexity of the scene.  This makes this likely to be extremely application dependent and the authors fail to recognize that in the discussion.  Please add text to address the point that while the selection of a 3 X 3 patch size makes sense for this application, it may not be optimal for all. A good example would be hyperspectral images of terrain acquired with modern sensors deployed on drones where the spatial resolution can be quite high but depending on the flight pattern large sections of the reconstructed image scene might be single components.  in this case a larger patch might be more appropriate.

-Line 631 makes reference to supplementary material summarizing the hyperparameters.  I could not locate any supplementary material for this paper.  If there is no supplementary material, this line should be removed. However, I do feel that information pertaining to the hyperparameters chosen would be good to include in the supplementary material.

- It was unclear from the introductory material and associated discussion whether the iterative training scheme was uniquely developed in this work, inspired from prior publications and modified for use here, or used as designed in a prior publication.  I suggest adding some discussion to clarify the novelty of the iterative method with respect to the application of classification of hyperspectral images.

- Line 127 contains a typo.  The word "mayrid" should be "myriad".

- Line 680 contains a typo.  The word "foe" should be "for"

Round 2

Reviewer 1 Report

Comments and Suggestions for Authors

All the isssues have been addressed. I think it can be published in the journal.

Reviewer 2 Report

Comments and Suggestions for Authors

The authors addressed my comments. I believe the paper is ready for publication.